# Large Language Models Are Zero-Shot Time Series Forecasters

**Nate Gruver**[*]
NYU

**Marc Finzi**[*]
CMU

**Shikai Qiu**[*]
NYU

**Andrew Gordon Wilson**
NYU

## Abstract

By encoding time series as a string of numerical digits, we can frame time series forecasting as next-token prediction in text. Developing this approach, we find that large language models (LLMs) such as GPT-3 and LLaMA-2 can surprisingly *zero-shot* extrapolate time series at a level comparable to or exceeding the performance of purpose-built time series models trained on the downstream tasks. To facilitate this performance, we propose procedures for effectively tokenizing time series data and converting discrete distributions over tokens into highly flexible densities over continuous values. We argue the success of LLMs for time series stems from their ability to naturally represent multimodal distributions, in conjunction with biases for simplicity, and repetition, which align with the salient features in many time series, such as repeated seasonal trends. We also show how LLMs can naturally handle missing data without imputation through non-numerical text, accommodate textual side information, and answer questions to help explain predictions. While we find that increasing model size generally improves performance on time series, we show GPT-4 can perform worse than GPT-3 because of how it tokenizes numbers, and poor uncertainty calibration, which is likely the result of alignment interventions such as RLHF.

## 1 Introduction

Despite similarities with other sequence modeling problems, such as text, audio, or video, time series has two particularly challenging properties. Unlike video or audio, which typically have consistent input scales and sampling rates, aggregated time series datasets often comprise sequences from radically different sources, sometimes with missing values. Moreover, common applications of time series forecasting, such as weather or financial data, require extrapolating from observations that contain a tiny fraction of the possible information, making accurate point predictions nearly impossible and uncertainty estimation especially important. While large-scale pretraining has become a key element of training large neural networks in vision and text, enabling performance to scale directly with data availability, pretraining is not typically used for time series modeling, where there is no consensus unsupervised objective and large, cohesive pretraining datasets are not readily available. Consequently, simple time series methods (e.g. ARIMA [8], and linear models [52]) often outperform deep learning methods on popular benchmarks [24].

In this paper, we demonstrate how large language models (LLM) can naturally bridge the gap between the simple biases of traditional methods and the complex representational learning and generative abilities of modern deep learning. In particular, we introduce an exceedingly simple method, **LLMTIME**[2], to apply pretrained LLMs for continuous time series prediction problems, illustrated at a high level in Figure 1. At its core, this method represents the time series as a string of numerical digits, and views time series forecasting as next-token prediction in text, unlocking

---

[*]Equal contribution
[2]https://github.com/ngruver/llmtime

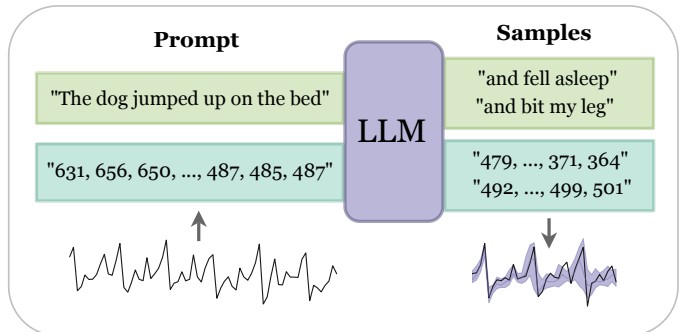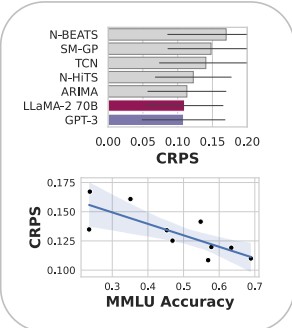

Figure 1: We propose **LLMTIME**, a method for time series forecasting with large language models (LLMs) by encoding numbers as text and sampling possible extrapolations as text completions. LLMTIME can outperform many popular time series methods without any training on the target dataset (i.e. zero-shot). The performance of LLMTIME also scales with the power of the underlying base model. Notably, models that undergo alignment (e.g. RLHF) do not follow the scaling trend. For example, GPT-4 demonstrates inferior performance to GPT-3 (Section 6).

the use of powerful pretrained models and probabilistic capacities, such as likelihood evaluation and sampling. To enable strong performance, we propose techniques to (1) effectively encode time series as a string of numerical digits and (2) adapt the discrete distributions of LLMs to continuous densities capable of modeling sophisticated multimodal distributions. Using these techniques, we find LLMTIME can exceed or match purpose-built time series methods over a range of different problems in a *zero-shot* fashion, meaning that LLMTIME can be used without any fine-tuning on the downstream data used by other models.

The zero-shot nature of LLMTIME carries several natural advantages: (1) it facilitates the straight-forward application of LLMs, eliminating the necessity for specialized knowledge of fine-tuning procedures and the substantial computational resources required for these procedures, as well as side-stepping access issues surrounding proprietary source code or APIs for LLM training or fine-tuning; (2) it is naturally suited to scenarios with limited data availability, where there is little information for training or fine-tuning; (3) by leveraging the broad pattern extrapolation capabilities of extensively pre-trained LLMs, it circumvents the extensive time, effort, and domain-specific expertise typically required for crafting dedicated time series models.

To understand the origins of LLMTIME's impressive performance, we investigate how LLMs express preferences for simple or repetitive sequences [19] and show that these biases are in fact compatible with the salient structure of time series, such as seasonality. Aside from these biases, LLMs also can naturally acccommodate missing data, and express multimodal distributions, which is particularly useful for time series. We also show how LLMs enable appealing functionality, such as the ability to provide additional side information through prompting, and query the LLM to explain its predictions.

Finally, in addition to broadly compelling forecasting performance, we find performance tends to improve with scale, and the quality of point predictions also improves with the quality of the uncertainty representation. However, we also find GPT-4 has worse uncertainty calibration than GPT-3, likely due to interventions such as reinforcement learning by human feedback (RLHF).

## 2  Background

**Language modeling**    Language models are trained on a collection of sequences, $\mathcal{U} = \{U_1, U_2, \ldots U_i, \ldots, U_N\}$, where $U_i = (u_1, u_2, \ldots, u_j, \ldots, u_{n_i})$ and each token, $u_i$, belongs to a vocabulary $\mathcal{V}$. Large language models typically encode an autoregressive distribution, in which the probability of each token is only dependent on the previous tokens in the sequence, $p_\theta(U_i) = \prod_{j=1}^{n_i} p_\theta(u_j \mid u_{0:j-1})$. The parameters, $\theta$, are learned by maximizing the probability of the entire dataset $p_\theta(\mathcal{U}) = \prod_{i=1}^{N} p_\theta(U_i)$. Every language model has an associated *tokenizer*, which breaks an input string into a sequence of tokens, each belonging to $\mathcal{V}$. Proper tokenization is extremely important, and small details can have surprisingly significant effects. The most common tokenization method for autoregressive language models is *byte-pair encoding* (BPE), which treats inputs like

bit strings and assigns tokens based on the rate of occurrence in the training corpus, optimizing for shorter sequences of tokens on average. Sampling from a language model typically starts with a *prompt*, $u_{0:k}$, and proceeds sequentially using $p_\theta (u_j \mid u_{0:j-1})$, which is often preprocessed, for example through temperature scaling or nucleus sampling [25].

**Large language models**    Brown et al. [9] showed that increasing a language model's parameter count and training data size leads to new capabilities such as *zero-shot generalization*, in which a model can perform a text-formatted task without training the model parameters on any task-specific data. Large language models, for example GPT-3 [9] or LLaMA-2 [44], accomplish this form of generalization through *in-context learning*, which identifies patterns in the language model's prompt and extrapolates them through next-token prediction. Many authors have speculated that in-context learning emerges from a language model's extensive compression of the input data [19, 42, 15]. Compression favors learning algorithms that operate over the input data with programmatic abstractions, for example, context-free grammars [2] or induction heads [34], which can implement copy-and-paste type operations for generating samples with highly structured syntax. In this work, we show that the zero-shot generalization abilities of LLMs and their preference for compressible patterns extend well beyond language understanding and can be used for time series forecasting.

Zero-shot generalization has made LLMs significantly more useful as assistants, leading to the create of methods to align LLMs with human preferences and instructions, for example reinforcement learning from human feedback (RLHF) [38] and instruction tuning [46]. While key to modern LLMs products, alignment methods can also significantly affect the abilities and calibration of the underlying model [36, 10]. Here we show these methods can also affect forecasting ability.

**Time series data**    Time series data typically takes the exact same form as language modeling data, as a collection of sequences $\{U_i = (u_1, \ldots, u_j, \ldots, u_{n_i})\}$, but in time series $u_j$ is numerical. Because language models are built to represent complex probability distributions over sequences, they are theoretically well-suited for time series modeling. In practice, however, language models are held back by the details of tokenizing numbers. BPE compresses numbers based on frequency of occurrence in the training data, so numbers can be broken down into awkward chunks that make learning basic numerical operations challenging. Touvron et al. [43] therefore designed the LLaMA tokenizer to map numbers to individual digits, which can lead to significant improvements in mathematical abilities, with small LLaMA models outperforming GPT-4 [30].

The other challenge of applying language models to time series data is proper evaluation. Mean absolute error (MAE) is commonly used but ignores uncertainty in the forecast which is highly limiting for stochastic data [24, 6]. Continuous ranked probability score (CRPS) captures distributional qualities and can compare models that generate samples without likelihoods. For a single prediction, the CRPS score is defined against the estimated cumulative distribution function (CDF), $\hat{F}$ as $\mathrm{CRPS}(\hat{F}, y) = \int_{\mathbb{R}} \left( \hat{F}(z) - \mathbb{I}_{(z-y)>0} \right)^2 dz$, where $\hat{F}(z)$ is the empirical CDF produced by sampling forecasts and $\mathbb{I}$ is the indicator function. While CRPS is an improvement over MAE, it also ignores key structures in the data, such as correlations between time steps. Fortunately, language models can assign likelihoods to full sequences of time series data, and we show how a small modification to an LLM's discrete likelihood can yield a continuous density that is useful for model comparison.

**Language models for time series**    Several authors have explored using pretrained language model encoders as initializations for time series models. For example, Zhou et al. [56] propose FPT, which finetunes a BERT encoder to perform time series forecasting. Similarly, Zhang et al. [53] introduce Meta-Transformer, a framework for finetuning a language model for non-text modalities, including time series. Fewer papers explore using LLMs as forecasters without finetuning. The only method we are aware of is PromptCast [50], which poses forecasting as question answering with prompting.

**Our work**    Unlike methods that leverage LLM backbones, our method is entirely zero-shot and does not require finetuning. Unlike PromptCast, we show that LLMs can be used directly as forecasters without any added text or prompt engineering, if we carefully preprocess the numerical values themselves. Our method solely relies on LLM's abilities to extrapolate patterns in general sequences and nothing particular to English or any other language. Going beyond prior work, we also cultivate the probabilistic nature of large language models and their ability to capture uncertainty over highly stochastic time series.

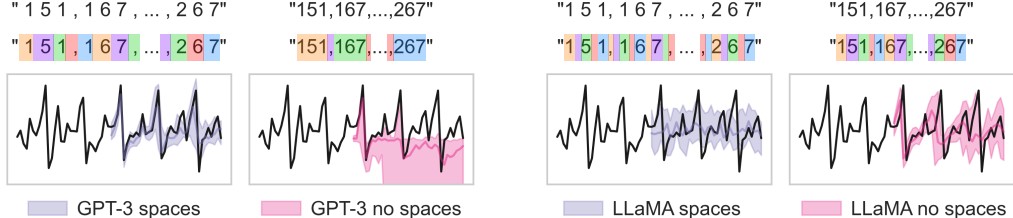

Figure 2: Careful tokenization is important for good forecasting with LLMs. Using the Australian Wine dataset from Darts [23], with values [151, 167, ..., 267], we show the tokenization used by GPT-3 [9] and LLaMA-2 [44] and the corresponding effect on forecasting performance. Added spaces allow GPT-3 to create one token per digit, leading to good performance. LLaMA-2, on the other hand, tokenizes digits individually, and adding spaces hurts performance.

## 3 LLMTIME: Forecasting with Language Models

Forecasting with LLMTIME has relatively few steps. Once the numerical values are processed into strings, making predictions with the language model follows standard sampling procedures. As we show next, however, correct pre-processing is not always intuitive but is extremely important, and incorrect handling can lead to unusable predictions.

**Tokenization** Tokenization is particularly important because it directly influences how patterns form within tokenized sequences and the types of operations that language models can learn. Unfortunately, common tokenization methods like BPE tend to break a single number into tokens that don't align with the digits, which can make arithmetic considerably more difficult [30]. For example, the number 42235630 gets tokenized as [422, 35, 630] by the GPT-3 tokenizer, and changes by even a single digit can result in an entirely different tokenization. By contrast, in many new open-source LLMs (e.g. LLaMA [43]), numbers are tokenized into individual digits by default. To remedy the tokenization of GPT models, we separate the digits with spaces to force a separate tokenization of each digit and use a comma (" ,") to separate each time step in a time series. Because decimal points are redundant given a fixed precision, we drop them in the encoding to save on context length. Thus, with e.g. 2 digits of precision, we pre-process a time series as follows before feeding into the tokenizer:

$$0.123, 1.23, 12.3, 123.0 \rightarrow \text{" 1 2 , 1 2 3 , 1 2 3 0 , 1 2 3 0 0".}$$

In Figure 2, we show that the added spaces of this encoding are helpful for GPT models, preventing the model from getting derailed by outputting an unusual token during sampling. For LLaMA models, with their unique tokenization of numbers, added spaces have the opposite effect. Each digit and space is already assigned its own token, and space tokens become nuisance inputs, adding to the sequence length without simplifying the sequence's structure and potentially making the sequence out-of-distribution to the model.

**Rescaling** To avoid wasting tokens when the inputs are very large, we scale values down so that the $\alpha$-percentile of rescaled time series values is 1. We avoid scaling by the maximum value so that the LLM can see some fraction of examples $(1 - \alpha)$ where the number of digits changes and reproduce this behavior in its outputs to produce larger values than it has seen. We also experiment with an offset $\beta$ based calculate as a percentile of the input data, and we tune these two parameters on validation log likelihoods (details in Appendix A).

**Sampling / Forecasting** To forecast, draw many samples (e.g. 20) from the LLM and use the statistics of the samples at each time step to construct a point estimate (e.g. as the median) or probabilistic forecast (e.g. as quantiles). To control sampling, we use temperature scaling, logit bias, and nucleus sampling (Appendix C).

**Continuous likelihoods** Modeling sequences of individual digits has additional benefits beyond good samples. With $n$ digits of precision in base $B$, each sequence of $n$ digits after the decimal place corresponds to one of $B^n$ possible bins (Figure 3), each with width $B^{-n}$. As each distribution $p(u_j \mid u_{0:j-1}; \theta)$ is a softmax over possible digits, we can view the distribution over each individual number as a hierarchical softmax [31], with $p(u_1, ..., u_n) = p(u_n|u_{n-1}, ..., u_0) p(u_1|u_0) p(u_0)$. Though a language model's probability distribution is discrete, we can easily adapt it to provide a

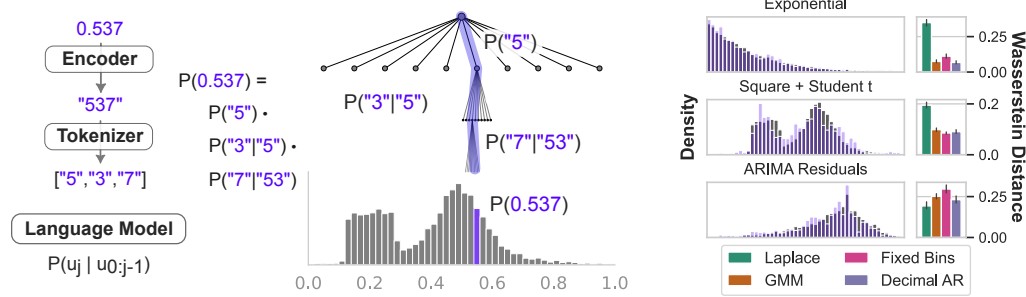

Figure 3: **Left:** Autoregressive models over sequences of digits act like hierarchical softmax distributions over the corresponding numbers. When combined with uniform distributions in each discrete bucket, distributions over strings can become expressive distributions over continuous domains. **Right:** Using simple autoregressive models (e.g. RNNs) trained on a string representation of numbers, we can fit complex distributions that can be challenging for other methods, such as heavy-tailed or multimodal distributions. A simple autoregressive model can match or outperform well-known methods for density estimation, such as Gaussian mixture models (GMMs) or binning with a fixed resolution, as measured by Wasserstein distance between samples.

continuous density by placing a uniform distribution in each bin. Enumerating each of the countably infinite numbers that the model can produce (because the model can output an arbitrary number of digits before the decimal point) with an index $k \in \mathbb{N}$ each with probability $p_k$, we can write out the distribution as a mixture of disjoint uniform distributions over the bins $p(x) = \sum_{k \in \mathbb{N}} p_k U_k(x)$ where $U_k(x) = B^n \mathbb{I}_{x \in [B^{-n}k, B^{-n}(k+1))}$. Therefore if a given data point lies in bin $k$, its continuous log likelihood is $\log p(x) = \log p_k + n \log B$. Finally, to obtain the likelihood $\log p(z)$ in the original input space, we add a change of variables factor $\log |\frac{dx}{dz}|$, where $z \mapsto x = s(z)$ is the rescaling operation in the pre-processing. As a result, the exponentially large number of bins and exponentially small bin widths enabled by our construction make it surprisingly efficient to represent flexible and high-resolution continuous distributions with LLMs, despite using a discrete tokenization of numbers.

**Language models as flexible distributions**     The fact that LLMs can express flexible distributions over numbers is key for time series data. Uncertainty quantification is essential to forecasting, and typical approaches to representing uncertainty in time series can be limited by misspecification. For example, one common method for creating a probabilistic forecast is to fit a Gaussian or Laplace observation model. When the underlying data distribution is multimodal, both of these models will perform poorly. Methods like Gaussian mixture models (GMMs) solve the issue of multimodality but introduce additional challenges to optimization and model selection. We show that a language model is an underrated solution by training a small autoregressive model on a variety of one-dimensional distributions shown in Figure 3 (right). These distributions come from an exponential random variable, a mixture of a uniform and a student-t distribution, and the heavy-tailed distribution of time series prediction residuals from an ARIMA model on the `MonthlyMilk` dataset [23]. We evaluate these fits quantitatively by computing Wasserstein distances, and compare to a Laplace observation model, a GMM trained with expectation-maximization, and logistic regression over a flat binning of the data (with a tuned bin size). Each model is trained with only 200 samples from the distribution. The results show that the decimal autoregressive language model ("Decimal AR") performs extremely well, handling asymmetric, multimodal, and heavy-tailed distributions, which are among the diverse types characteristic of time series data.

## 4    Experiments

We evaluate the zero-shot forecasting ability of LLMs by comparing LLMTIME with GPT-3 and LLaMA-2 70B to many popular time series baselines on a variety of benchmark time series datasets. Not only is LLMTIME able to generate plausible completions of the real and synthetic time series, it achieves higher likelihoods and CRPS values in zero-shot evaluation than the dedicated time series models like ARIMA, TCNs, and N-HiTS. When evaluated on deterministic metrics like MAE, LLMs also perform well, obtain the best or second best MAE values on each benchmark. As we are using

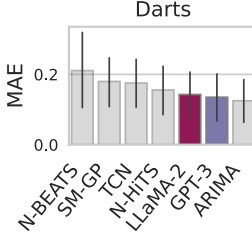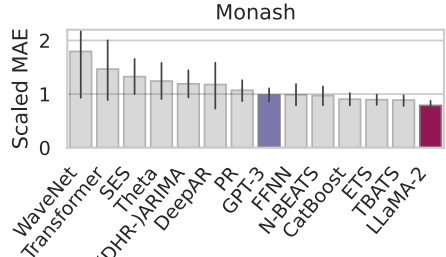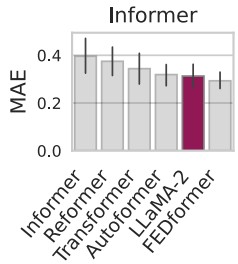

Figure 4: LLMTIME with base model GPT-3 or LLaMA-2 70B has the best or second best aggregated performance on several deterministic time series benchmarks [23, 18, 54] while being entirely zero-shot. Collectively, these benchmarks comprise 29 individual datasets with diverse sources, lengths, and noise levels. For Monash MAE numbers, established results are reported on unnormalized data, so we normalize values before aggregating (Appendix C.2). The informer datasets are multivariate, and we predict each covariate independently with LLMTIME (Appendix C.3). GPT-3 evaluation on the Informer datasets was skipped because of the cost of API queries. Error bars show standard errors over the individual datasets in each benchmark.

LLMs with undisclosed datasets, data leakage is an important concern that we address directly in Appendix B. Beyond strong performance on standard benchmarks, which are the most useful for comparison, we find that LLMTIME also performs well on datasets that could not have been present in the base model's training data. The full set of hyperparameters used for LLMTIME and the baseline methods are detailed in Appendix C.1. For some of the longer time series, not all of the history can be fit into the context window, and hence hyperparameters implicitly capture the trade-off between higher precision and capturing a larger temporal history.

**Datasets**   We use three benchmark datasets that are common within deep learning research and many baseline methods that accompany the benchmark datasets.

- **Darts** [23]: A collection of 8 real univariate time series datasets. For Darts, we use several methods that are implemented directly in the package, including neural network models (TCN [28], N-BEATS [37], N-HiTS [11]) and simple moving average models (ARIMA [8]). Darts enables learning observation models with tractable likelihoods and is therefore especially useful for benchmarking the probabilistic predictions of LLMTIME. We also include Spectral Mixture Gaussian Process (SM-GP) [48], a Bayesian nonparametric method (details in Appendix C.1).

- **Monash** [18]: The Monash forecasting archive contains 30 publicly available datasets along with baseline numbers for 12 forecasting models, including simple exponential smooth (e.g. ETS [26]), gradient boosting (e.g. CatBoost [39]) and deep learning models (e.g. DeepAR [40], WaveNet [35]). The Monash archive comprises over 400,000 individual time series, making it infeasible to use in its entirety with the largest available LLMs. To reduce the computational burden, we evaluate GPT-3's zero-shot performance on 19 datasets described in Appendix C.2.

- **Informer** [54]: We evaluated on multivariate datasets widely used for benchmarking efficient transformer models [16, 54]. In order to predict multivariate data with LLMTIME, we forecast each covariate independently. We baseline against numbers obtained by running public implementations from the Autoformer [49] and FEDFormer [55] codebases (Appendix C.3).

**Deterministic results**   To compute MAE values for LLMTIME we use the pointwise median of 20 samples from the base model (GPT-2 or LLaMA-2 70B). Figure 4 shows that deterministic predictions from LLMTIME are ranked best or second best on all the considered benchmarks while having no trainable parameters. We provide visualizations of the forecasts in Appendix C.5/C.7/C.8.

**Probabilistic results**   In Figure 5, we show several probabilistic evaluations on the Darts datasets, including aggregated NLL and CRPS numbers, as well as analysis of how each model reacts to decreasing the input data size. Evaluated on log likelihood and CRPS, LLMTIME considerably outperforms the baselines in aggregate and on almost every individual dataset (results per dataset included in Appendix C.5). Given the analysis of language model-derived densities in Section 3, it is unsurprising that language models excel in probabilistic evaluations, outperforming the baselines even

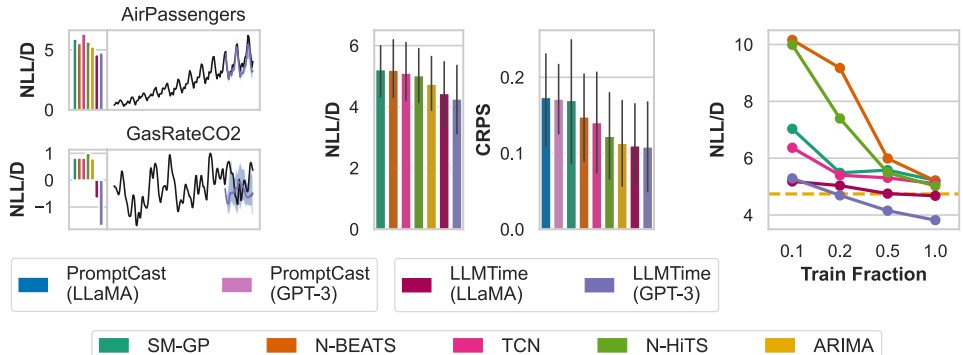

Figure 5: Extended experiments on the Darts datasets. **Left:** Example probabilistic forecasts with baseline negative log likelihood per dimension (NLL/D). LLMs easily extrapolate trends (e.g. AirPassengers) and reproduce local patterns when data is noisy (e.g. GasRateCO2). **Center:** When using probabilistic metrics like NLL and CRPS, LLMTIME outperforms all baselines, including PromptCast [50], a competing LLM method. Error bars show standard errors over datasets with Darts. **Right:** LLMTIME is much more sample efficient than competing methods. While the performance of other methods degrades rapidly when we restrict them to a fraction of the original training set, LLMTIME can assign high likelihood with only a few examples.

more dramatically. In Figure 5 (left) we show two informative examples that capture the performance of LLMTIME. When extrapolating the AirPassengers dataset, LLMTIME successfully identifies and continues trend and period components, with uncertainty that grows as predictions get further from the input data. On GasRateCO2, LLMTIME replicates local structure when there is relatively little global structure. In Figure 5 (right) we show that LLMTIME not only performs better than baselines with access to the full training data but also when restricted to small fractions of the training data. As time series is frequently characterized by relative data scarcity and challenges in transfer learning, the data efficiency of LLMs is especially attractive.

**Comparison with PromptCast**     Though included in the results described above, we want to explicitly highlight that LLMTIME significantly outperforms PromptCast [50] when applied to both GPT-3 and LLaMA-2 70B, according to CRPS and MAE aggregated over the Darts datasets. This performance gap highlights important differences between the two approaches. Unlike our method, PromptCast formulates forecasting as a conventional question-answering task in NLP by prompting pre-trained language models with an explicit question about future values in a time series. For example, PromptCast feeds in the prompt "The values in the WoolyDataset for the past 95 time steps are 6172, 6709, 6633, . . . , 6077. What will the values for the next 5 time steps be? The values for the next 5 time steps will be", to extract predictions from an LLM. PromptCast also does not apply our tokenization and data rescaling strategy (Section 3), which we show is crucial for good performance.

## 5   Origins of Zero-Shot Performance

To understand why LLMs can extrapolate time series in a zero-shot manner, let's take a step back and consider simple numerical sequences, for example $[1, 4, 9, 16, . . . ]$ or $[0, 0, 1, 0, 1, 2, 0, 1, 2, 3, . . . ]$. For any input sequence, there are arbitrarily many generation rules that are consistent with the input (e.g. $f(x) = x^2$ for $x \in [1, 2, 3, 4, ...]$), but some generation rules are overly complex and will generalize poorly. LLMs can forecast effectively because they prefer completions derived from simple rules, adopting a form of Occam's razor prior [19, 42, 15]. To explicitly demonstrate this phenomenon, we create a synthetic example using the function $f(x) = x + \cos(x)$ with additive Gaussian noise. We fit symbolic expressions to the first 70% of timesteps using PySR [14] with symbols ["+", "·", "-", "/", "sin", "cos", "exp","square"] to identify generating rules with known complexity, quantified by the number of symbols in the regressed expression (Appendix D). Figure 6 (left) shows the likelihood that GPT-3 assigns the highest likelihood to symbolic regression generating rules that balance consistency with complexity.

In Figure 6 (right) we show how program induction in LLMs leads to good zero-shot prediction for many deterministic patterns common in time series data. Along with samples, we also show

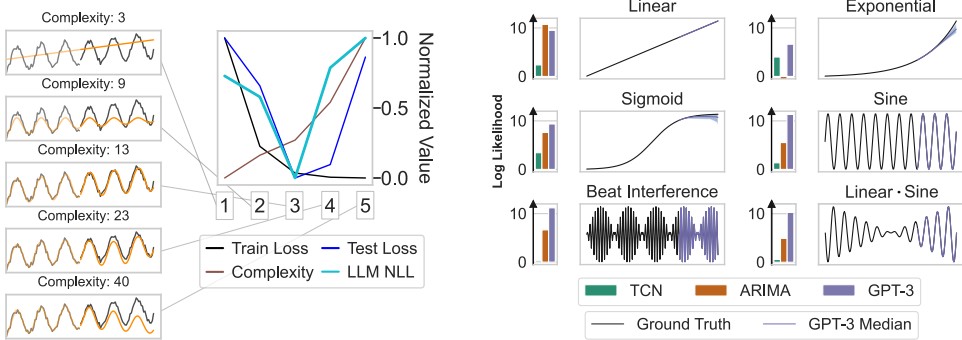

Figure 6: LLMs can find low complexity explanations of the data, enabling them to zero-shot extrapolate numerical sequences. **Left:** GPT-3 likelihoods favor solutions from symbolic regression (PySR [14]) that balance training loss and complexity, leading to good generalization. **Right:** GPT-3 predicted median and 10-90th percentile prediction interval are shown given 140 timesteps of context. On the right of each time series, we show the log likelihoods compared to the ARIMA and TCN time series models. Overall, GPT-3 performs considerably better than the baselines, though composition and exponential growth are more challenging for the models (Appendix D.1).

likelihoods, comparing against standard time series models, which often struggle to extrapolate these simple patterns because they cannot identify a programmatic generation rule to make predictions unlike those seen in the observed history. While the generic simplicity bias is helpful for identifying and extrapolating patterns in the input, a number of patterns common in time series models also translate directly to known capabilities of language models, for example

- **Repetition bias and periodicity**: LLMs' bias towards repetitive sequences [25] (often unwanted in NLP) corresponds precisely to the ability to identify and extrapolate periodic structure in the input. $4.2, 8.6, 1.0, 4.2, 8.6$ will lead to a $1.0$ as a likely next output without any time series or arithmetic knowledge ($x_t = x_{t-T}$).

- **Arithmetic and trend components**: LLMs' ability to perform addition and multiplication [51, 30] maps on to extrapolating linear and exponential trends. For example, predicting the next element of $0.2, 1.6, 3., 4.4$ the LLM needs only to add $1.4$ to the last element ($x_{t+1} = x_t + c$). Similarly, exponential trends have the generating rule $x_{t+1} = c \cdot x_t$ and sigmoid trends have the generating rule $x_{t+1} = x_t + cx_t(1 - x_t)$.

Combining multiple patterns together presents a more difficult challenge, as it requires both identifying the composite pattern and being able to perform the multiple operations within the same token budget. Supposing that a model can perform copying in a single forward pass and addition in a single forward pass, that does not necessarily mean that it can do both simultaneously. We find that GPT-3 is only sometimes able to perform these compositions, though GPT-4 does so more consistently as shown in Appendix E. It is likely that the limitations on compute and tokens spent may make this composition unnecessarily hard, and that additional recursive structure, for example from a scratchpad [33], Chain of Thought (CoT) prompting [47], or adaptive computation [41, 3], would make this task easier.

## 6 Special Properties of LLMs

So far we've shown that LLMs are effective forecasters across a variety of datasets and that their forecasting ability arises from biases created by generative pretraining. LLMTIME offers a mechanism for large-scale pre-training that is uncommon in machine learning for time series. LLMs lessen the amount of time series data that must be aggregated for pretraining, substituting text pretraining in its place, and enable more powerful scaling results. Beyond escaping the limits of task-specific data, text pretraining also has many test-time benefits that stem from the base model's ability to process and generate natural language. As we show in the following section, LLMs can leverage their abilities in order to seamlessly incorporate missing data or answer questions about time series.

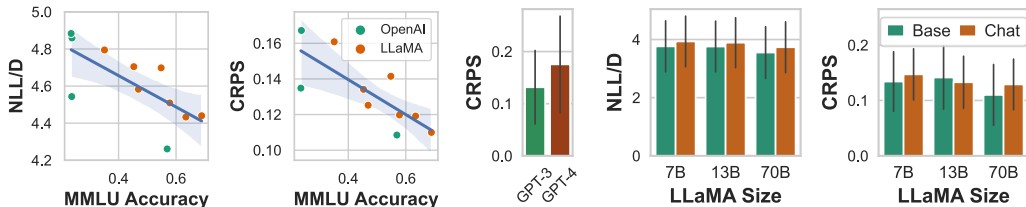

Figure 7: **Left:** Time series forecasting performance (NLL/D and CRPS on Darts [23]) improves with reasoning performance of the underlying model LLM, as judged by accuracy on the Massive Multitask Language Understanding (MMLU) benchmark [22]. Displayed results are for all GPT-3, LLaMA [43], and LLaMA-2 [44] base models. **Center:** GPT-4 performs worse than GPT-3. **Right:** Forecasting performance (NLL/D and CRPS on Darts) appears to be negatively affected by alignment procedures (e.g. instruction tuning and RLHF) in general. LLaMA-2 chat models typically perform worse than the corresponding base model. Error bars show standard errors over individual datasets.

**Base models and forecasting performance**   Given the rapid growth and improvement in open-source LLMs [43, 44], the relationship between LLMTIME forecasting performance and the performance of the underlying base model is particular important and has broad implications. Steady increases in LLM benchmark performance can directly translate to steady improvements in forecasting ability. In Figure 7 (right), we show a study with OpenAI models (davinci, babbage, curie, ada), variants of LLaMA [43] (7B, 13B, 33B, 65B) and LLaMA-2 [44] models (7B, 13B, 70B) measuring accuracy on the Massive Multitask Language Understanding (MMLU) benchmark and probabilistic forecasting error. As we might hope, when reasoning (MMLU) performance increases forecasts also improve.

**Chat models**   Though convenient scaling relationships appear to hold for base models, the correlation begins to break down when we consider models that have been post-processed for chatbot applications. GPT-4 [36], for example, demonstrates considerably more intelligence than GPT-3 and LLaMA models in natural language tasks, but effectively applying it to time series is challenging. In Figure 7 (center), we show that GPT-4 has a forecasting error (CRPS) significantly larger than GPT-3's on the Darts datasets. The performance drop is the result of several small details in GPT-4's method. Due to the altered tokenization, GPT-4 cannot be easily forced to tokenize individual digits into an unbroken stream of numbers. Due to the restrictions on the API, likelihood evaluations are also not permitted, which is why we present results for only CRPS. While GPT-4 can perform well on the synthetic examples discussed in Section 5 (shown in Appendix E), we find that the GPT-4 calibration is much worse than GPT-3's on stochastic data, likely as a result of the preprocessing details above and the fact that the model has been treated with RLHF [13] which is known to degrade calibration on question-answering tasks [36]. GPT-4 is not the only example of degraded performance in models designed for chat functionality. We observed the same phenomenon in LLaMA-2 models, which have corresponding chat versions for every model size. Figure 7 (right) shows that chat versions tend to have markedly worse forecasting error than their non-chat counterparts, though still maintain trends in size and reasoning ability.

**Missing data**   A major advantage of treating forecasting as a text completion task and using LLMs is that we can easily feed in any input that can be encoded as text. Often in time series, the time series will be incomplete and certain observations are missing. Simple imputation methods, such as nearest neighbor, are still core pre-processing steps in common data science workflows [32], and the choice of imputation method is especially relevant to clinical data, which often contains irregularly sampled measurements and where missingness can be meaningful signal in itself [27]. Much like humans reading partial reports, LLMs can handle missing values without imputation by adopting special symbols, for instance,

$$[64, , , 49, , 16, ] \rightarrow \text{"64, NaN, NaN, 49, NaN, 16, NaN"}$$

. In Figure 8 we compare likelihoods and CRPS value for forecasts from traditional time series methods and LLaMA-2 70B on data that has been corrupted with missing values and then processed with linear interpolation and the above string formatting. While the likelihoods of traditional methods rapidly deteriorate with corruptions, we find that LLaMA-2 70B is more resilient, and when comparing CRPS values, LLaMA-2 70B is competitive with methods that use interpolation.

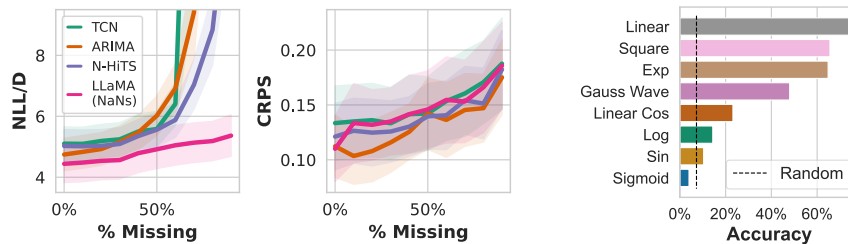

Figure 8: **Left:** LLMTIME can handle missing values without interpolation by denoting missingness with text (e.g. 'NaN'). For baseline methods we perform linear interpolation and then fit the model as usual. LLMTIME assigns higher log likelihood to datasets preprocessed with added 'NaN's than baseline methods assign to interpolated datasets. Forecasting performance, as judged by CRPS, is competitive between LLMTIME and alternative methods that use explicit interpolation. Filled area shows standard error over individual datasets and 3 random seeds. **Right:** LLMs can be used to answer questions about time series data posed as text. We show GPT-4's accuracy at predicting the function that generated the time series, obtained using chain-of-thought prompting.

**Connecting time series and textual understanding** Because LLMs are designed for natural language and code, we can augment the numerical time series with useful text. We can do so either by providing textual side information as inputs, or by producing textual outputs from a given time series. An interesting question is whether GPT-4 can explain in text its understanding of a given time series. We probe this quality by providing GPT-4 the code to generate our synthetic time series, provide the values of one these time series, and then ask it to infer which of the functions produced the data in a zero-shot manner. The prediction accuracies are shown in Figure 8, with the three remaining rows all being 0. With CoT [47] prompting the model performs much better than random chance; however, its ability to identify patterns better when directly extrapolating the numerical data, suggesting that its numerical understanding is not fully connected to its textual understanding. In making predictions, the model often explains properties of the time series in order to select the right candidate from the list, and we show several of these sample explanations in Appendix F. We also show how this task is encapsulated in a simple (unprompted) next token prediction problem on cells of a Jupyter notebook, illustrating why we expect such capabilities to emerge with a sufficiently powerful language model.

## 7 Discussion

We have demonstrated that large language models can be used as pretrained time series forecasters by encoding numerical values as text. As with other "foundation" models, pretraining confers useful biases toward generalizable patterns that would otherwise be engineered into the model through architecture design [21], and enables natural scaling of performance with improvements in the base pretrained model. Because LLM forecasters are trained on language, they also confer unconventional capabilities, such as question answering. More broadly, framing time series forecasting as natural language generation can be seen as another step towards unifying more capabilities within a single large and powerful model, in which understanding can be shared between many tasks and modalities. Moreover, *zero-shot* forecasting can enable broadly compelling performance without requiring significant computational resources, domain expertise, or many downstream training data points.

While LLM forecasters benefit from the strengths of pretrained transformers, they also inherit their weaknesses, which can include a limited context window. While many univariate time series problems can fit comfortably within increasingly large context windows, multivariate problems pose a more significant challenge. There have been several recent advances extending LLM context windows to 10-100K tokens [36, 4, 5, 1]. Combining these advances with time series forecasting is a particularly exciting direction for future research. Another potential challenge of using current LLMs architectures could be their weakness in arithmetic and performing recursive and compositional operations, which could be a limitation on particularly challenging time series. On the other hand, many time series do not require precise arithmetic. Understanding the extent to which this is the case, and relaxing this limitation, is also a promising avenue for future research. Separately from any limitation, it would also be promising to investigate effective procedures for fine-tuning LLMs on time series. We hope that bridging LLM research with time series forecasting brings benefits to both communities.

**Acknowledgements.** We thank Micah Goldblum, Greg Benton, and Wesley Maddox for helpful discussions. This work is supported by NSF CAREER IIS-2145492, NSF I-DISRE 193471, NSF IIS-1910266, BigHat Biosciences, Capital One, and an Amazon Research Award.

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
