# Appendix

## Table of Contents

## A  Detailed method and hyperparameters

### A.1  Input scaling

For all baseline methods, we use the MinMaxScaler from sklearn. For GPT-3, since it can handle inputs spanning multiple orders of magnitudes by using varying number of digits, we apply an affine transformation to each element $x_t$ of a time series $(x_1, ..., x_T)$: $x_t \mapsto (x_t - b)/a$, where $b = \min_t x_t - \beta(\max_t x_t - \min_t x_t)$, and $a$ is the $\alpha$-percentile of the shifted series $(x_1 - b, ..., x_T - b)$. We also consider a basic scaler that only applies scaling and not shifting, with $a$ clipped to a maximum of $0.01$ when the series only has tiny values. Here $\alpha$ and $\beta$ are hyperparameters controlling the thresholds at which the number of digits used by the language model changes.

### A.2  Validation tuning

We construct a validation time series from the last $T$ observations in the training series, where $T$ is the length of the test series. When the training series is shorter than $2T$, we take the last half of the training series as the validation series. The likelihood of generating the validation conditioned on the remaining training series is used to select the hyperparameters. Since LLMTIME is zero-shot, the likelihood is computed without training. For other methods such as ARIMA, the likelihood is computed after training on the remaining training series.

## B  Addressing Memorization Concerns in GPT-3 Evaluations

Evaluating the performance of black box APIs, like those provided by OpenAI, can be challenging when training data for the underlying models is unknown. In our time series setting, it is natural to

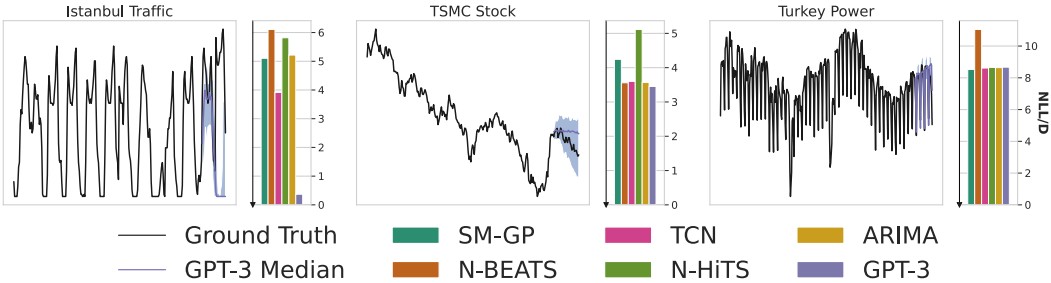

Figure 9: Evaluation on a collection of short univariate time series recorded after GPT-3's training cutoff date. We compare the performance of our GPT-3 predictor against popular time series models. Predicted median and 10-90th percentile intervals are shown for GPT-3 given the context, and we compare test negative log likelihoods. GPT-3 continues to be competitive with or outperforms the baselines on all of the tasks, from in-context learning alone. This result reinforces our belief that GPT-3's performance is not due to memorization of the test data.

wonder if the common benchmark datasets we use are included in the GPT-3 training data. LLMs are known to memorize large amounts of their training data verbatim, including common benchmark text datasets and copyrighted material [7, 12]. Beyond outright memorization, more benign data leakage of closely related data is also possible, leading to overestimation of the generalization performance.

Even if our evaluation datasets are present in the GPT-3 training data, it's unlikely that GPT-3's good performance is the result of memorization for at least two reasons a priori. First of all, our idiosyncratic formatting is unlikely to be present in the training dataset, even if the numerical values and their order are the same. Second, the time series datasets are unlikely to appear in GPT-3's training data sufficiently frequently to lead to memorization, as memorization increases in proportion with redundancy [29].

To further address the memorization concern, we also perform a direct experiment to show GPT-3 also demonstrates strong performance when evaluated on time series recorded after its training data cutoff date, September 2021. We use the following 3 time series:

- Istanbul Traffic (source: https://www.kaggle.com/datasets/leonardo00/istanbul-traffic-index): This dataset provides minute-by-minute Traffic Index data for Istanbul from October 2022 to May 2023. We select the "TI" column and downsample the series to an hourly frequency for the period from May 5th, 2023 to May 18th, 2023, resulting in a total of 267 observations.

- TSMC Stock (source: https://www.kaggle.com/datasets/yeemeitsang/tsmc-stock-exchange-2022): This dataset contains daily stock market trading data for Taiwan Semiconductor Manufacturing Company Limited for the year 2022. We use the closing price column, which consists of a total of 246 observations.

- Turkey Power (source: https://www.kaggle.com/datasets/dharanikra/electrical-power-demand-in-turkey): This dataset includes hourly electricity generation and consumption data for Turkey from January 1st, 2020 to December 31st, 2022. We choose the "Total" column and downsample to daily data for the year 2022, resulting in 366 observations.

For each time series, we reserve the last 30 observations as test data and perform hyperparameter tuning for each method over the same grid as in Appendix C.1. As displayed in Figure 9, GPT-3 not only predicts plausible continuations of each time series but also competes with or even surpasses the performance of the baseline models in all the tasks, solely based on in-context learning. This result reinforces our belief that GPT-3's performance is not due to memorization of the test data.

# C Benchmarking details and extended results

## C.1 Darts datasets

For the Darts datasets, we use the GPyTorch library [17] for Gaussian Process implementation and the DARTS libary [23] for ARIMA, TCN, N-BEATS, N-HiTS. We use default values for hyperparameters not described below. The test set is the last $20\%$ of each series.

We use several baseline methods implemented directly in Darts [23]:

- **ARIMA**: ARIMA [8], short for AutoRegressive Integrated Moving Average, has been a popular choice for time series forecasting for many decades.
- **TCN**: Temporal Convolutional Network (TCN) [28] is residual network with dilated 1D convolutions.
- **N-BEATS**: N-BEATS [37] is a deep learning model tailored for time series forecasting. It employs a deep architecture with backward and forward residual links and stacked fully-connected layers.
- **N-HiTS**: N-HiTS [11] is a deep learning model that incorporates hierarchical interpolation and multi-rate data sampling techniques in order to create forecasts that emphasize different frequencies and scales of the input signal.

We also include Spectral Mixture Gaussian Process (**SM-GP**) [48] as a Bayesian nonparametric approach to time series modeling.

We include the exact hyperparameters for each method below:

**GPT-3** We perform a grid search over $\alpha \in [0.5, .7, 0.9, 0.99], \beta \in [0, .15, 0.3, .5]$, precision (number of decimals) $\in [2, 3]$, and $\mathrm{temperature} = 0.7$.

**GPT-4** Since likelihood evaluation is not available for GPT-4, we fix its hyperparameters for all datasets as follows: we use the basic scaler with $\alpha = 0.3$ and $\mathrm{temperature} = 1.0$ with $\mathrm{top\ p} = 0.8$. We do not insert spaces between digits for GPT-4 since it uses a different tokenizer than GPT-3 for which this strategy is not effective.

**LLaMA** For models LLaMA-1 (7B/13B/30B/70B) and LLaMA-2 (7B/7B-chat/13B/13B-chat), we perform a grid search over $\mathrm{temperature} \in [0.2, 0.4, 0.6, 0.8]$ and use $\alpha = 0.99, \beta = 0.3, \mathrm{precision} = 3, \mathrm{nucleus} = 0.9$. For LLaMA-2 70B and LLaMA-2 70B-chat we use $\mathrm{temperature} = 1.0, \alpha = 0.99, \beta = 0.3, \mathrm{precision} = 3, \mathrm{nucleus} = 0.9$.

**Spectral Mixture Gaussian Process (SM-GP)** We use a GP with a kernel formed by the sum of a spectral mixture kernel with 12 mixture components and a RBF kernel. We tune the learning rate from [5e-3, 1e-2, 5e-2, 1e-1].

**ARIMA** We perform a grid search over $p \in [12, 20, 30], d \in [1, 2]$, and $q \in [0, 1, 2]$.

**TCN** We perform a grid search over input_chunk_length $\in [10, 100, 400]$, output_chunck_length $\in [1, 10]$, kernel_size $\in [3, 5]$, num_filters $\in [1, 3]$, and likelihood $\in$ [Laplace, Gaussian].

**N-BEATS** We perform a grid search over input_chunk_length $\in [10, 100, 400]$, output_chunk_length $\in [1, 10]$, layer_widths $\in [64, 16]$, num_layers $\in [1, 2]$, and likelihood $\in$ [Laplace, Gaussian].

**N-HiTS** We perform a grid search over input_chunk_length $\in [10, 100, 400]$, output_chunck_length $\in [1, 10]$, layer_widths $\in [64, 16]$, num_layers $\in [1, 2]$, and likelihood $\in$ [Laplace, Gaussian].

## C.2 Monash datasets

We evaluate on 19 datasets in Monash that satisfy two criteria

1. The total number of individual series cannot be prohibitively large, so that the experiments can be run in time without access to an enormous cluster and without a gratuitous API expenses.

2. The length of the forecasting horizon cannot extend to a length that makes it impossible to fit both the forecast and the history into the context window of the language model.

When we applied these criteria, we obtained the following 19 datasets were selected: covid deaths, solar weekly, tourism monthly, australian electricity demand, pedestrian counts, traffic hourly, hospital, fred md, tourism yearly, tourism quarterly, us births, nn5 weekly, nn5 daily, traffic weekly, saugeenday, cif 2016, bitcoin, sunspot.

To aggregate across datasets, we normalized the mean absolute error by the MAE achieved by simply predicting the last observed value before the test series (a naive baseline). This normalization places high weight on datasets for which methods perform significantly better or worse than the naive predictor.

Several of the baseline methods in the archive are shared with Darts, and all descriptions and code can be found in [18]. A few notable addition include

- **CatBoost**: CatBoost [39] is gradient-boosting framework for continuous or categorical data.
- **FFNN**: A feed-forward neural network with a fixed window of input and output, inspired by Goodfellow et al. [20].
- **PR**: A linear pooled regression (PR) model proposed by Trapero et al. [45].

We include visualizations of GPT-3's prediction on these datasets in Appendix C.7.

**GPT-3 hyperparameters**  We use the following hyperparameters for GPT-3: $\alpha = 0.9, \beta = 0$, temperature $= 0.7$. To avoid exceeding the context window, we truncate the history to at most 500 most recent observations. For the baselines, we report their performance as presented in [18]. The normalized MAE values shown in Figure 4 (center) are obtained by normalizing by the lowest baseline MAE on each dataset before aggregating.

**LLaMA-2 70B hyperparameters**  We use the following hyperparameters for LLaMA-2 70B: $\alpha = 0.99, \beta = 0.3$, temperature $= 1.0$, nucleus $= 0.9$. To avoid exceeding the context window, we truncate the history to fit in the LLaMA-2 context window (4096 tokens).

### C.3  Informer datasets

There are 6 datasets used by Zhou et al. [54] that have become standard benchmarks for evaluating efficient transformers. We evaluate on the 5 datasets that are typically used with a prediction horizon of 96 or 192: "ETTm2", "exchange_rate", "electricity", "traffic", and "weather". The results provided in the main text are for a prediction horizon of 96, and we include results for prediction horizon 192 in Appendix C.6. To make evaluation tractable with LLMTIME, we use a smaller evaluation set for each dataset, taking the last 96 or 192 timesteps of each series within each dataset as the test set. As there are many individual series in each multivariate dataset, the number of individual timesteps in the test sets is still substantial. To forecast multivariate series with LLMTIME we simply forecast over each series independently, combine the results, and evaluate as in prior work. Our efficient transformer baselines include

- **Informer**: Informer [54] is an efficient transformer model with sparse attention designed for long sequences.
- **Reformer**: Reformer [54] uses a locality-sensitive hashing mechanism to improve the memory use of attention.
- **Autoformer**: Autoformer [49] is a model design for long time series that replaces standard attention with a mechanism in Fourier space.
- **FEDformer**: Like Autoformer, FEDformer [55] uses frequency-based decompositions to construct an efficient alternative to attention.

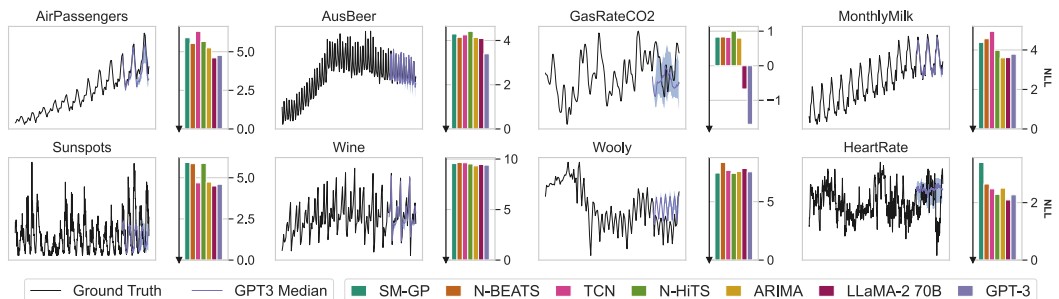

Figure 10: Median predictions of LLMTIME (GPT-3) and NLLs from LLMTIME (GPT-3 and LLaMA-2 70B) for every dataset within Darts [23]. The shaded area shows the 10th to 90th quantiles of the distribution over samples. LLMTIME consistently obtains better likelihood values than the baselines and often makes surprisingly accurate forecasts by effectively extrapolating trend and periodic components.

**LLaMA-2 70B hyperparameters** We use LLaMA-2 70B with $\alpha = 0.99, \beta = 0.3$, temperature $= 1.0$, nucleus $= 0.9$, precision $= 3$. The series in the Informer datasets are very long and we put as much as possible in the LLaMA-2 context window (4096).

## C.4 Synthetic datasets

For the baselines, we use the same hyperparameter grid in Section C.1. For GPT-3, we didn't find it useful to perform validation tuning. We use the basic scaler with $\alpha = 0.1$ and temperature $= 0.7$.

## C.5 Darts full probabilistic results

In Figure 10 we show the predicted NLLs and forecasts from LLMTIME using GPT-3 and LLaMA-2 70B as base models. LLMTIME typically obtains much better likelihoods than baselines and successfully identifies trend and seasonal components in the time series. We attribute this strong performance in part to the fact that the time series are relatively short. With the tokenization of the input, only about 300 of the observations can fit into the context window, and among the datasets only `Sunspots` and `HeartRate` exceed this amount (with 705 and 900 observations respectively).

## C.6 Informer datasets with extended horizon

Figure 11 shows MAE results per dataset and in aggregate for the Informer datasets we used in the paper. Extending the results in the main text, we also include MAE numbers for a prediction horizon of 192. We observed a similar trend overall, though the relative performance of LLMTIME is slightly diminished, largely due to the "electricity" and "traffic" datasets.

## C.7 Monash dataset visualizations

Figure 12 shows visualizations of the LLMTIME's median predictions (GPT-3 base model) on a subset of the Monash datasets.

## C.8 Informer dataset visualizations

Figure 13 shows visualizations of the LLMTIME's median predictions (LLaMA-2 70B base model) on the Informer datasets, for a subset of the each set of multivariate series.

# D Simplicity bias experiments

We generate data from the function $f(x) = \cos(x) + x$ and add Gaussian noise with zero mean and variance $0.05$. We fit symbolic expressions to the first 140 timesteps using PySR [14] with symbols ["+", "·", "-", "/", "sin", "cos", "exp","square"] and maxsize $= 70$, maxdepth $=$

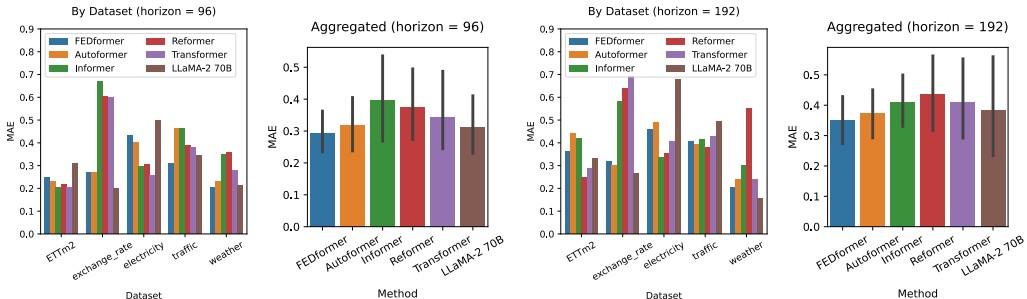

Figure 11: Aggregated and non-aggregated MAE numbers for LLMTIME (LLaMA-2 70B base model) and baselines on the Informer datasets. Overall LLMTIME performs well in aggregate for a zero-shot method, but its performance is highly variable, being the best method on some datasets and the worst on others. The relative performance of LLMTIME is slightly diminished for a longer prediction horizon, but LLMTIME is still very competitive with the best methods in aggregate. Error bars show two standard deviations in the error over datasets.

$10, \text{population\_size} = 50, \text{loss} = \text{abs}(\text{prediction} - \text{target}), \text{model\_selection} = \text{accuracy}$ and $\text{niterations} = 100$. The solutions are saved and ranked by complexity, which is simply the number of terms in the symbolic regression function. The five solutions shown in Figure 6 are

1. $(x_0 + 0.3652524)$
2. $\cos(\cos(x_0 / - 0.031412385) * (-1.5252972 + x_0))$
3. $(\sin(\cos(\cos(x_0/0.031470172) * -1.4668792)) + (\cos(0.81965065) * x_0))$
4. $(\sin(\cos(\cos((x_0/\sin(-0.03127001)) + 0.07646165) * -1.4539052)) + (\sin(\sin(\cos(\cos(\exp(\cos(-0.03127001) + x_0)))))) * x_0))$
5. $(\cos((\cos((x_0/ - 0.03127001) + 0.07646165)/ - 0.957405)/\sin(\sin(\cos(x0 - x0)) * \exp(\cos(\sin(x_0/ - 0.983214)))))/(\cos(\sin(\sin(\sin(\sin(x_0)) - (x_0 * (-0.47036648 - (x_0/0.5857117)))))) - -0.10476875))$

To obtain likelihoods we run GPT-3 ('text-davinci-003') with $\text{alpha} = 0.99, \text{beta} = 0.3, \text{basic} = \text{True}, \text{precision} = 1, \text{signed} = \text{True}$.

### D.1 Full synthetic predictions

Figure 14 shows likelihoods and forecasts from LLMTIME with GPT-3 on the full set of synthetic datasets. We see that some compositional tasks like Linear + Cosine are challenging, while others (Linear * Sine or X * Sine) are well within the abilities of the model. As shown above, GPT-3 demonstrates good understanding of Linear + Cosine through its likelihoods, but has more trouble in sampling. This discrepancy could be the result of good solutions being high likelihood while not being *typical*.

## E  GPT-4

We investigated using GPT-4 for time series prediction. Due to the limitations of the tokenizer, we used the naive tokenization strategy of feeding in the numbers without additional spaces. In addition, due to the enforced separation between system and user in the interface (through additional tokens we cannot modify), inputting the time series input alone leads GPT-4 to talk about the time series or provide analysis, rather than simply continuing the stream of numbers. In order to coax GPT-4 to produce numerical predictions which can be decoded, we added the additional commands System: "You are a helpful assistant that performs time series predictions. The user will provide a sequence and you will predict the remaining sequence. The sequence is represented by decimal strings separated by commas." User: "Please continue the following sequence without producing any additional text. Do not say anything like 'the next terms in the sequence are', just return the numbers. Sequence:". We found that doing so was sufficient to be able to consistently decode the output numerically for GPT-4, but not for GPT-3.5-turbo.

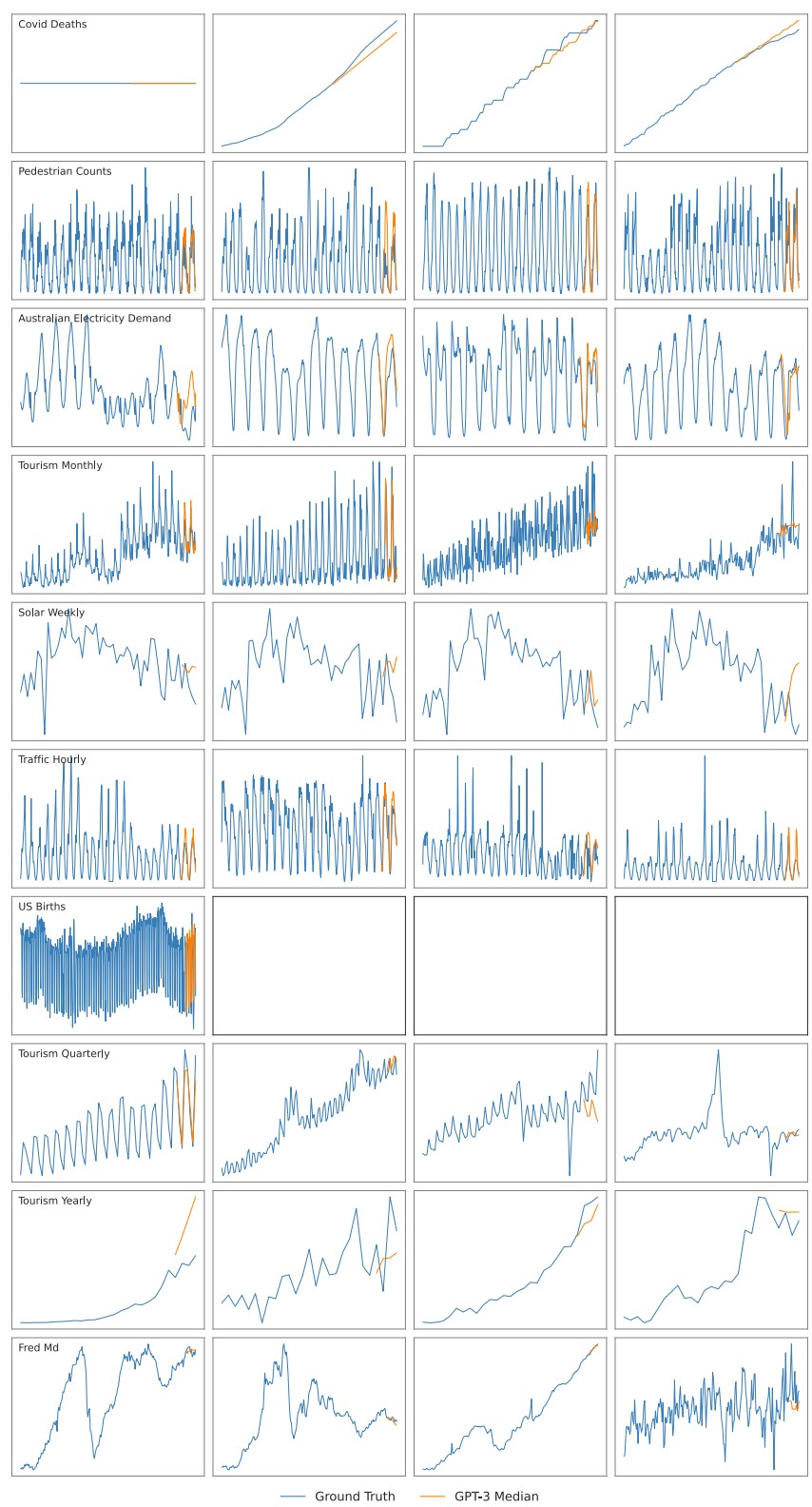

Figure 12: LLMTIME (GPT-3 base model) median predictions on at most 4 randomly chosen series per Monash dataset.

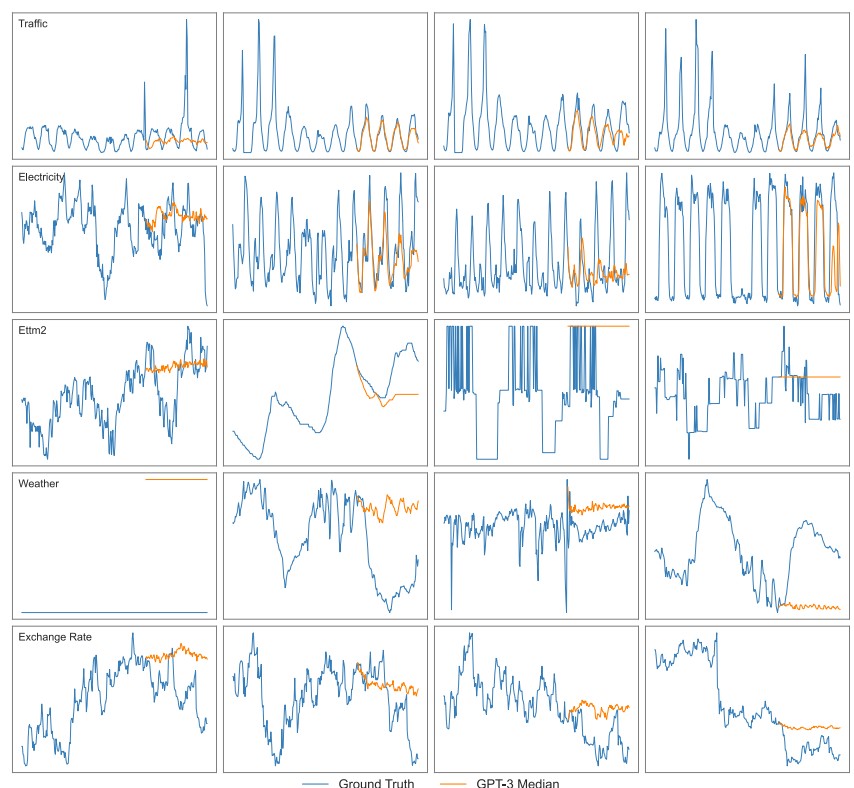

Figure 13: LLMTIME (LLaMA-2 70B base model) median predictions on 4 randomly chosen series per Monash dataset.

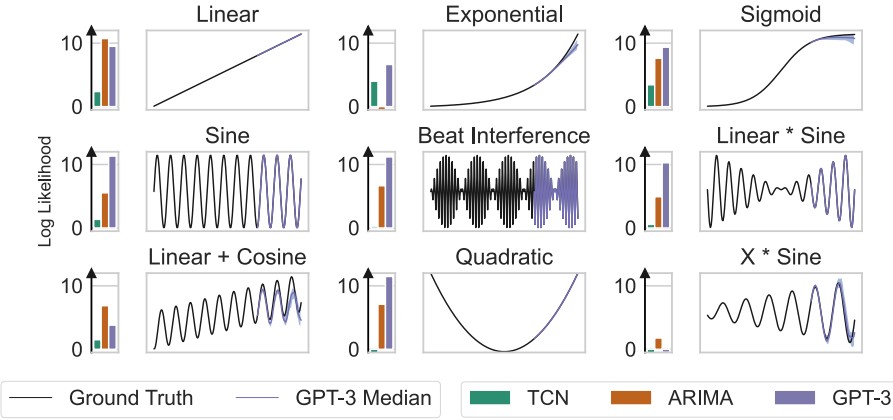

Figure 14: LLMTIME median predictions on all synthetic datasets using GPT-3 as a base model. The hyperparameters used are described in Appendix C.4.

We show predictions on the synthetic benchmarks (from Figure 6) in Figure 15. As one can observe, GPT-4 is considerably better performing on these synthetic benchmarks, although numerical decoding of the model sometimes fails before the full output. With non-deterministic time series problems such as with the DARTS datasets, the predictions are slightly worse than GPT-3, but the uncertainties are much less well calibrated as shown in Figure 16.

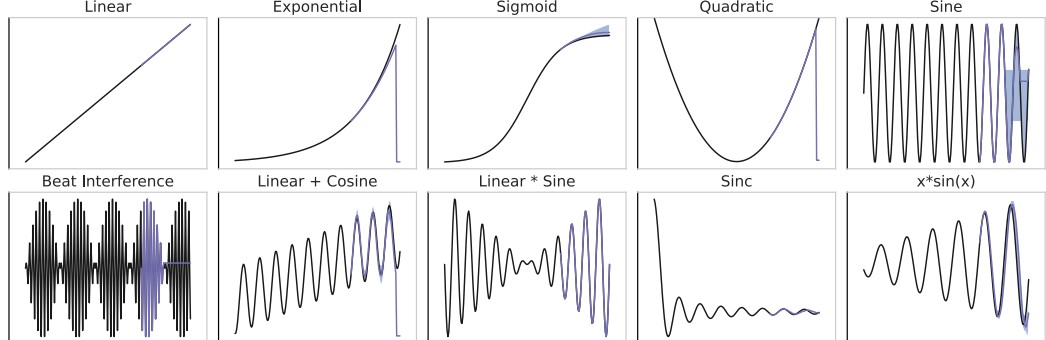

Figure 15: GPT-4 extrapolations on synthetic data (10-90th percentiles shaded). GPT-4 is able to identify and extrapolate the pattern for each of the deterministic time series, but sometimes behaves erratically.

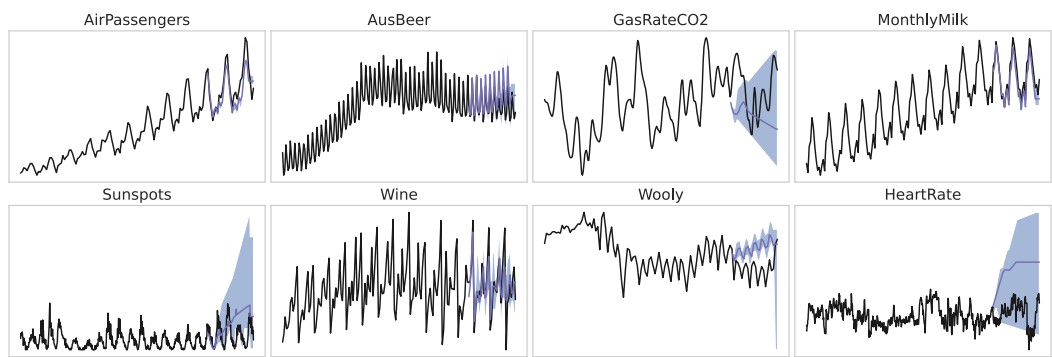

Figure 16: GPT-4 extrapolations on real (DARTS) time series (10-90th percentiles shaded). The extrapolations are plausible but worse than GPT-3, and the uncertainties tend to be more poorly calibrated making for a high CRPS.

# F   Multimodal Text Understanding of Time Series

We evaluate the ability of the language model to reason about the input time series through text in a zero-shot fashion. To test this, we devise a simple experiment where we generate a synthetic time series from one of several candidate functions. We provide the generation code and the numerical values to GPT-4 (Listing 1), but because of the randomness, GPT-4 must infer which of the functions generated the values. We note that as this code could easily be found within a Jupyter notebook on the internet without intentionally being designed as an experiment for LLMs, we should expect that this textual time series identification task will fall within the data distribution, and in principle should be solved given sufficient capabilities of the language model.

To make the problem slightly easier, we add an additional guiding prompt before and after the text in Listing 1. We prepend

> "The following is code that was run to generate a synthetic time series. From the input and output you will be asked to identify which of the time series was picked to generate the data."

to the code, and after the time series we append either

> "Which name gave rise to this series? Put your answer in the form 'Answer: gaussian_wave' "

or

"Carefully analyze the time series. Think step by step, make observations about the time series that you see and then use your observations to identify which of the functions is most likely to have generated it. Reason your way to a solution and at the end give give a name as your answer such as 'Answer: gaussian_wave'."

for chain-of-thought prompting.

The prediction accuracies computed over 20 trials are shown in Figure 8, with `x_times_sine`, `beat`, and `sinc` not shown in the table because GPT-4 predicted these incorrectly $100\%$ of the time. With the CoT prompting, this prediction task elicits some interesting textual analysis of the time series. Several (non cherry-picked) examples are shown below. Notably, this task elicits the model to analyze the time series in text, reasoning about the trend and periodicity. However, the model sometimes makes incorrect deductions about the behavior of the data it has seen, or the expected behavior of the candidate functions.

```
1  import numpy as np
2  mapping = {
3    'gaussian_wave': lambda t: np.exp(-5*(t-.6)**2)*np.sin(20*(t-6)),
4    'exp':lambda t: np.exp(2*t),
5    'linear_cos':lambda t: 0.3+ 0.5*t +.2*np.cos(25*t+3),
6    'linear':lambda t: 0.3+ 0.5*t,
7    'sine':lambda t: np.sin(40*t+3),
8    'sinc': lambda t: np.sin(10*t)/t/10,
9    'beat': lambda t: np.sin(3*t)*np.sin(25*t),
10   'sigmoid': lambda t: 1/(1+np.exp(-4*t)),
11   'log': lambda t: np.log(1+t),
12   'x_times_sine': lambda t: 4*(t+1)*np.sin(10*(t+1)+4),
13   'square': lambda t: 3*(t-.6)**2,
14 }
15 name = np.random.choice(list(mapping.keys()))
16 t = np.linspace(-1,1,200)+.1*np.random.randn(1)
17 x = mapping[name](t)
18 np.set_printoptions(formatter={'float': lambda x: "{0:0.3f}".format(x)
       })
19 print("Series: ",x)
20 print(" ",name)
21
22 Series:
23 [-0.000 -0.033 -0.070 -0.111 -0.153 -0.197 -0.240 -0.281
24 -0.320 -0.355 -0.385 -0.408 -0.425 -0.433 -0.432 -0.422
25 -0.402 -0.371 -0.330 -0.279 -0.217 -0.145 -0.064 0.026
26  0.124 0.229 0.339 0.453 0.570 0.688 0.806 0.922 1.033
27  1.140 1.238 1.328 1.407 1.474 1.527 1.564 1.586 1.590
28  1.576 1.543 1.491 1.420 1.329 1.219 1.091 0.945 0.782
29  0.604 0.413 0.209 -0.005 -0.227 -0.455 -0.686 -0.917
30  -1.147 -1.373 -1.591 -1.799 -1.995 -2.175 -2.338 -2.481
31  -2.602 -2.698 -2.769 -2.812 -2.826 -2.811 -2.766 -2.689
32  -2.583 -2.446 -2.279 -2.084 -1.863 -1.615 -1.345 -1.053
33  -0.743 -0.418 -0.081 0.265 0.617 0.970 1.322 1.667 2.003
34  2.325 2.630 2.915 3.175 3.407 3.609 3.778 3.910 4.005
35  4.060 4.074 4.046 3.975 3.862 3.706 3.509 3.272 2.997
36  2.686 2.341 1.965 1.563 1.137 0.692 0.232 -0.239 -0.715
37  -1.192 -1.665 -2.128 -2.577 -3.008 -3.414 -3.791 -4.136
38  -4.443 -4.710 -4.932 -5.108 -5.233 -5.306 -5.326 -5.291
39  -5.202 -5.058 -4.859 -4.608 -4.306 -3.955 -3.558 -3.120
40  -2.643 -2.132 -1.592 -1.029 -0.447 0.148 0.749 1.350
41  1.946 2.530 3.095 3.637 4.148 4.623 5.057 5.445 5.782
42  6.064 6.288 6.450 6.548 6.580 6.546 6.444 6.275 6.039
43  5.740 5.378 4.956 4.479 3.951 3.376 2.760 2.109 1.429
44  0.726 0.009 -0.717 -1.444 -2.164 -2.870 -3.555 -4.210
45  -4.830 -5.408 -5.936 -6.409 -6.822 -7.170 -7.448 -7.653
```

```
46    -7.783 -7.834 -7.806 -7.698 -7.511 -7.245]
```

Listing 1: Self-contained code presented to the model for the multimodal time series identification task. When the code is run one of the listed functions is randomly chosen to generate the time series. In order to simply predict the next token after observing this text (which could be found in the cells of a Jupyter notebook) the model must infer which of the functions produced the series.

```
1  First, let's analyze the time series and make some observations:
2
3  1. The time series starts at a negative value, increases linearly, and
      ends at a positive value.
4  2. There is no apparent periodicity or oscillation in the time series.
5  3. There are no sudden jumps or changes in the time series, indicating
      a smooth function.
6
7  Now, let's go through the functions in the mapping and see which one
      matches our observations:
8
9  - 'gaussian_wave': This function has a periodic oscillation, which
      doesn't match our observations.
10 - 'exp': This function increases exponentially, which doesn't match
      the linear increase in our observations.
11 - 'linear_cos': This function has a periodic oscillation, which doesn'
      t match our observations.
12 - 'linear': This function increases linearly, which matches our
      observations.
13 - 'sine': This function has a periodic oscillation, which doesn't
      match our observations.
14 - 'sinc': This function has a periodic oscillation, which doesn't
      match our observations.
15 - 'beat': This function has a periodic oscillation, which doesn't
      match our observations.
16 - 'sigmoid': This function has a sigmoid shape, which doesn't match
      our observations.
17 - 'log': This function increases logarithmically, which doesn't match
      the linear increase in our observations.
18 - 'x_times_sine': This function has a periodic oscillation, which
      doesn't match our observations.
19 - 'square': This function has a parabolic shape, which doesn't match
      our observations.
20
21 Based on our observations and analysis, the function that most likely
      generated the time series is 'linear'.
22
23 Answer: linear
```

Listing 2: Sample output for Linear time series (correctly classified)

```
1  First, let's analyze the time series.
2
3  The time series starts with a value of 0.148 and ends with a value of
       8.063. This indicates that the series is monotonically increasing.
4
5  Now, let's look at the differences between consecutive values. The
       differences are small and seem to be increasing as the series
       progresses. This suggests that the series is not only increasing
       but also accelerating.
6
7  With these observations, let's analyze the given functions:
8
9  1. gaussian_wave: This function represents a wave that is localized
       around a specific point. It does not match the characteristics of
       the time series.
10 2. exp: This function represents an exponential growth, which matches
       the increasing and accelerating characteristics of the time series
       .
11 3. linear_cos: This function represents a linear function with an
       added cosine term. It does not match the accelerating
       characteristic of the time series.
12 4. linear: This function represents a linear growth, which does not
       match the accelerating characteristic of the time series.
13 5. sine: This function represents a sinusoidal wave, which does not
       match the characteristics of the time series.
14 6. sinc: This function represents a sinc function, which does not
       match the characteristics of the time series.
15 7. beat: This function represents a beat pattern, which does not match
        the characteristics of the time series.
16 8. sigmoid: This function represents a sigmoid curve, which does not
       match the characteristics of the time series.
17 9. log: This function represents a logarithmic growth, which does not
       match the accelerating characteristic of the time series.
18 10. x_times_sine: This function represents a product of a linear
       function and a sinusoidal function, which does not match the
       characteristics of the time series.
19 11. square: This function represents a quadratic growth, which does
       not match the characteristics of the time series.
20
21 Based on our analysis, the exp function is the most likely candidate
       to have generated the time series.
22
23 Answer: exp
```

Listing 3: Sample output for Exp time series (correctly classified)

```
1  Observations about the time series:
2  1. The values are positive and continuous.
3  2. The values start from a high value and gradually decrease until
        they reach a minimum value near the middle of the series.
4  3. After the minimum value, the values gradually increase again until
        they reach a high value near the end of the series.
5  4. The overall trend of the time series seems to be symmetric.
6
7  Now, let's analyze the given functions and their characteristics:
8
9  1. gaussian_wave: Symmetric, bell-shaped curve, values are positive
        and continuous.
10 2. exp: Exponential growth, not symmetric.
11 3. linear_cos: Linear trend with oscillations, not symmetric.
12 4. linear: Linear growth, not symmetric.
13 5. sine: Oscillating function, not symmetric.
14 6. sinc: Symmetric, sinc function has a peak at the center and
        oscillates on both sides.
15 7. beat: Combination of two sine waves with different frequencies, not
         symmetric.
16 8. sigmoid: S-shaped curve, not symmetric.
17 9. log: Logarithmic growth, not symmetric.
18 10. x_times_sine: Product of a linear function and sine function, not
        symmetric.
19 11. square: Quadratic function, symmetric, parabolic shape.
20
21 Based on our observations, the most likely function to have generated
        this time series is the 'square' function. It is symmetric, has a
        parabolic shape, and the values are positive and continuous.
22
23 Answer: square
```

Listing 4: Sample output for Square time series (correctly classified)

```
1  We can start by analyzing the time series and making observations
        about its characteristics:
2
3  1. The time series starts and ends at 0, with values close to 0 in the
        beginning and end.
4  2. The time series has a single peak and a single valley, reaching a
        maximum value around 0.85 and a minimum value around -1.
5  3. The time series is not periodic, as it doesn't repeat its pattern.
6
7  Now, let's go through the functions one by one and see which one
        matches these observations:
8
9  - 'gaussian_wave': This function is a product of a Gaussian and a
        sinusoidal function, which could give us a single peak and valley.
        However, it is periodic, so it doesn't match observation 3.
10 - 'exp': This function is an exponential function, which doesn't match
        the observations as it doesn't have a peak and valley.
11 - 'linear_cos': This function is a linear function combined with a
        cosine function, which could give a single peak and valley.
        However, it is also periodic, so it doesn't match observation 3.
12 - 'linear': This function is a linear function, which doesn't have a
        peak and valley, so it doesn't match observation 2.
13 - 'sine': This function is a sinusoidal function, which is periodic
        and doesn't match observation 3.
14 - 'sinc': This function is a sinc function, which has multiple peaks
        and valleys, so it doesn't match observation 2.
15 - 'beat': This function is a product of two sinusoidal functions,
        which could give a single peak and valley. However, it is also
        periodic, so it doesn't match observation 3.
16 - 'sigmoid': This function is a sigmoid function, which matches all
        the observations. It starts and ends at 0, has a single peak and
        valley, and is not periodic.
17 - 'log': This function is a logarithmic function, which doesn't have a
        peak and valley, so it doesn't match observation 2.
18 - 'x_times_sine': This function is a product of a linear function and
        a sinusoidal function, which could give a single peak and valley.
        However, it is also periodic, so it doesn't match observation 3.
19 - 'square': This function is a quadratic function, which doesn't have
        a peak and valley, so it doesn't match observation 2.
20
21 Based on these observations, the most likely function to have
        generated the time series is 'sigmoid'.
22
23 Answer: sigmoid
```

Listing 5: Sample output for Gaussian_wave time series (misclassified)

```
1  Let's analyze the time series step by step:
2
3  1. The time series starts at a negative value, increases to a positive
        value, and then decreases back to a negative value. This suggests
        that the function might have a symmetric or periodic behavior.
4
5  2. The time series has a peak in the middle, which indicates that the
        function might have a single maximum or minimum point.
6
7  3. The time series does not show an exponential growth or decay, which
        rules out 'exp', 'log', and 'sigmoid' functions.
8
9  4. The time series does not have a linear pattern, which rules out the
        'linear' function.
10
11 5. The time series does not show a clear periodic pattern, which rules
        out 'sine', 'linear_cos', and 'beat' functions.
12
13 6. The time series does not show a sinc pattern, which rules out the '
        sinc' function.
14
15 7. The time series does not show a clear x_times_sine pattern, which
        rules out the 'x_times_sine' function.
16
17 8. The time series does not show a clear parabolic pattern, which
        rules out the 'square' function.
18
19 Based on the above observations, the only remaining function is '
        gaussian_wave'. Therefore, the most likely function to have
        generated the time series is:
20
21 Answer: gaussian_wave
```

Listing 6: Sample output for Linear_Cos time series (misclassified)