# OpenReview forum: "Large Language Models Are Zero-Shot Time Series Forecasters"
_NeurIPS.cc/2023/Conference — NeurIPS 2023 poster_

### Official Review · Reviewer_kUEg · 2023-07-02

**Soundness:** 2 fair
**Presentation:** 3 good
**Contribution:** 2 fair
**Rating:** 4
**Confidence:** 3

**Summary:**

The paper discusses a novel approach for time series forecasting by encoding time series as numerical digits and treating it as next-token prediction in text. Surprisingly, it is found that large language models (LLMs) like GPT-3 can effectively extrapolate time series even without being specifically trained for this task. The success of LLMs in time series forecasting is attributed to their ability to extrapolate deterministic patterns, such as seasonality, and model uncertainty by using expressive distributions on continuous variables. It is argued that LLMs excel in predicting numerical sequences due to biases introduced by their text pre-training, which favors simple explanations. Additionally, LLMs demonstrate excellent representation of multimodal distributions by decomposing the distribution over digits. This combination of qualities allows LLMs to achieve strong performance on time series data and enable new capabilities such as integrating non-numerical text and handling missing data. Furthermore, the study shows that the performance of LLMs in time series prediction improves with increased scale, suggesting significant potential for this unconventional approach.

**Strengths:**

1. The paper proposed a method that utilize the pretrained language model to solve time series forecasting problems.
2. This paper proposes a novel timing coding approach that can help LLMS understand timing data.
3. Also, the authors identify why language models are able to perform well time-series prediction task.

**Weaknesses:**

1. The experimental results are not enough to support the viewpoint put forward in the paper. For example, most experiments are conducted on virtual data sets, but how are the effects on real data sets is not enough.

2. For some of the methods proposed in this paper, their effectiveness has not been fully analyzed, and there is a lack of analysis of Tokenization, scale, etc.

3.As can be seen from the results in Figure 4, the proposed method is effective only for some simple time series data (such as obvious periodicity or trend), but not for some more complex time series data (such as wooly or HeartRate).

**Questions:**

1. The contributions are not very clear, and it seems that the authors are only utilized the GPT models.

2. The authors claim they explain why LLMs can be used for timing prediction, but there is not enough analysis in the paper.

3. Why are most experiments conducted on virtual datasets and not on more real-world datasets?

**Limitations:**

See weakness and questions.

---

> ### Author Rebuttal · Authors · 2023-08-10
>
> Thank you for your comments. While we appreciate your feedback, we believe that based on the concerns you raise, some of the experiments we performed may have been overlooked.
>
> **"most experiments are conducted on virtual datasets"**
> We do not believe this is an accurate assessment. For time series extrapolation, we have performed comparisons on $8$ real datasets from the Darts project (shown in figure 6, and figure 4), the $10$ most impactful real datasets from the Monash archive (figure 5), and $3$ additional real world datasets collected after the GPT-3 cutoff date (figure 9 and appendix C). With the figure 1 in the attached PDF of the general comment, we have extended these further with an additional $5$ autoformer datasets. This is in comparison to the $8$ synthetic datasets that we evaluate in figure 3.
>
> **"For some of the methods proposed, there effectiveness has not been fully analyzed, and there is a lack of analysis of Tokenization, scale, etc"**
> In section 4 and figure 4 we analyze precisely these design details, considering tokenization with and without spaces and varying the base of the encoding of the numbers. We evaluate these choices both qualitatively and quantitatively.
>
> **"... proposed method is effective only for some simple time series data"**
> We have evaluated on a large variety of real time series datasets and on these datasets our method consistently ranks in the top performing methods. Keep in mind that we are showing the median and 10-90th percentiles, which for time series with a high variance will look quite different from realizations of the time series or samples from the model. On Wooly ours is indeed not the best performing method, but on HeartRate due to the considerable variance in the underlying process our method achieves the lowest NLLs among the competing methods.
>
> **"authors are only using GPT-3 models"**
> We would like to point to figure 7 (right) in the main text where we compare against publicly accessible Cohere, Forefront, and Aleph-Alpha models as the base LLM used in our method (with additional details in appendix B.3). Since the reviews, we have made an effort to extend this even further, with the powerful LLaMA-2 models. We have added additional comparisons using LLaMA-2 70B as the base LLM in the response PDF figures (1,2) and the other sized LLaMA-2 models in figure (3).
>
> Altogether we hope that in light of these clarifications about experiments in the text and additional ones that we have run, that you may consider reevaluating the paper.

---

> > ### Comment · Reviewer_kUEg · 2023-08-19
> > **Thank you for your response**
> >
> > I would like to first thank the author for their informative and active response, especially the additional experiments on some other forecasting tasks. After reading the author response, I found some of my concerns have been addressed.
> >
> > But one question still remains that, the ability of the LLM is a mystery, why would that perform forecasting with such performance in zero-shot manner? I agree with Reviewer AFnc that the author seemed not active on discovering how this happened and was only showcasing their great performance on simple experiment settings. Another thing I am wondering that, the reasoning ability is lack for LLMs, how could it perform this task in tokenized space?
> >
> > As a result, I'm very curious about the experiment details and the results in your rebuttal response PDF, i.e., the forecasting evaluation with Autoformer, etc. What are the detailed experimental settings such as the multi-run number, the evaluation protocol including datasets, forecasting horizons, evaluation methods, etc.
> > What are the detailed results of all the compared methods? On what datasets? Significance test? I know it is not recommended but I really would like to see the case studies on how they forecast in those long real-world datasets.
> >
> > I think it is an interesting paper. I really believe that the author would give a thorough analysis about the above discussion. But from the view of the current version, I think it lacks the experiments on the existing SoTA works like PatchTST, Autoformer, TimesNet on simple datasets, and lacks the detailed discussion on complicated datasets such as those in author's general response part. It also lacks deep analysis on the mystery of LLMs on these numerical tasks.
> >
> > I am inclined to remain my rating.

---

> > > ### Author Response · Authors · 2023-08-21
> > > **Follow-up clarifications (part 1)**
> > >
> > > Thank you for engaging in the rebuttal discussion!
> > >
> > > **Why LLMs can perform forecasting**
> > >
> > > We understand why you are surprised by the strong zero shot performance of LLMs. We were also surprised. As we explore in the submission, however, the ability of LLMs is not a complete mystery. We show that LLMs perform well because of two key properties:
> > >
> > > 1. LLMs prefer simple sequences over more complex sequences. For numerical sequences generated from a simple algorithm (e.g. a linear or exponential function) LLMs can perform zero-shot extrapolation because completions that are consistent with pattern in the past observations are more simple than completions that are not consistent with the pattern. Therefore LLMs’ preference for simple sequences leads to good extrapolations. Because time series data often consists of noise observations of a deterministic exponential (e.g. viral spread) or periodic (e.g. weather) trend, ability to extrapolate basic numerical sequences also leads to good forecasting performance.
> > >
> > > 2. LLMs can represent uncertainty over continuous variables. Time series data is often very noisy, and therefore a good time series forecaster must be good at both extrapolating deterministic patterns and estimating the distribution over possible outcomes. We show that LLMs implicitly represent a hierarchical softmax distribution over the input space by outputting a distribution over each digit in a number. It is able to represent complex distributions (e.g. with multiple modes or heavy tails) more effectively than several popular alternatives, as we show in Figure 2 of the submission.
> > >
> > > It is therefore not a complete mystery why LLMs can perform well. Just as Autoformer can perform well on time series data because it has inductive biases that are well-suited to time series data (e.g. extrapolating periodic features with Fourier decompositions), we show that LLMs have inductive biases that are also well-suited to time series data (e.g. a bias towards simple patterns and flexible uncertainty representation). Compared with many other papers that propose time series architectures, for example PatchTST, which you mention in your comment, we actually provide more explanation for the performance of our method. PatchTST is a vanilla Transformer model trained on time series broken into independent channels and chunks of those channels [1]. It has no biases specifically designed for time series (as the Transformer was originally designed for NLP), and it is therefore not clear *a priori* why it performs so much better than alternative approaches. Yet, like LLMs, PatchTST consistently works well.
> > >
> > > [1] Nie, Yuqi, et al. "A time series is worth 64 words: Long-term forecasting with transformers." arXiv preprint arXiv:2211.14730 (2022).

---

> > > > ### Author Response · Authors · 2023-08-21
> > > > **Follow-up clarifications (part 2)**
> > > >
> > > > **Experimental details**
> > > >
> > > > We are happy to provide more of the experimental details for the autoformer comparison (though many of the relevant details are also provided in the rebuttal PDF). To obtain the numbers for the baseline methods, we follow the exact code and procedure in the original Autoformer paper (and later FEDformer paper). You can find the exact performance of every method and every dataset in the rebuttal PDF. Below are the datasets, forecasting horizons, and evaluation metrics we used:
> > > >
> > > > - Datasets: ETTm2, Exchange, Electricity, Traffic, Weather.
> > > > - Forecasting horizons: 96, 192
> > > > - Evaluation metrics: MAE
> > > >
> > > > We did not include a significance test because we were not aware of any standard practice for significance testing in the Autoformer or FEDformer paper. The only difference between our evaluation and the evaluation in the original papers is the precise test set. We use the last 96 or 192 time steps of each dataset as the test set in order to make comparison with LLaMA feasible, as there are hundreds of individual time series (one per covariate) for each dataset.
> > > >
> > > > Unfortunately, it’s not possible to include additional plots for LLaMA’s predictions at this point in the discussion period, but we will include such plots in a camera ready version of the paper. Qualitatively, the predictions of LLaMA on the Informer datasets mirror the predictions from GPT-3 shown in Figure 6 of the submission. On periodic datasets, the model is able to identify the period and extrapolate the trend. On noisy datasets, the samples approximate the local patterns of past observations and the median of the samples extends the high-level trend (i.e. increasing, decreasing, or constant values).
> > > >
> > > > **Additional baselines**
> > > >
> > > > While a comparison with Autoformer, TimesNet, and PatchTST on DARTS would also be a good addition to our paper, we would like to note that our existing baselines perform on-par with TimesNet and significantly better than Autoformer on established short-horizon forecasting benchmarks. The following numbers are taken directly from the TimesNet paper [1] (page 7), which compares methods on the M4 datasets:
> > > >
> > > > | Model | SMAPE | MASE | OWA |
> > > > |:-------------|---------:|---------:|---------:|
> > > > | TimesNet | 11.829 | 1.585 | 0.851 |
> > > > | N-HiTS | 11.927 | 1.613 | 0.861 |
> > > > | N-BEATS | 11.851 | 1.599 | 0.855 |
> > > > | FEDformer | 12.840 | 1.701 | 0.918 |
> > > > | Autoformer | 12.909 | 1.711 | 0.939 |
> > > >
> > > > NHiTS and NBEATS have almost the exact same performance as TimesNet and much better performance than Autoformer overall and are therefore clearly strong baselines. Likewise, we can compare MAE numbers on the informer datasets using data from the NHiTS paper [2] (numbers for NHiTS, NBEATS) and PatchTST paper [3] (numbers for  PatchTST/64, FEDformer, and Autoformer):
> > > >
> > > > | Model | Weather (96) | Weather (192) | Traffic (96) | Traffic (192)  | ILI (96) | ILI (192) | ETTm2 (96) | ETTm2 (192) | Electricity (96) | Electricity (192) |
> > > > |:-------------|---------:|---------:|---------:|---------:|---------:|---------:|---------:|---------:|---------:|---------:|
> > > > | PatchTST/64 | 0.198 | 0.241 | 0.249 | 0.256 | 0.754 | 0.870 | 0.256 | 0.296 | 0.222 | 0.240 |
> > > > | N-HiTS | 0.195 | 0.247 | 0.282 | 0.297 | 0.869 | 0.934 | 0.255 | 0.305 | 0.249 | 0.269 |
> > > > | N-BEATS | 0.203 | 0.261 | 0.282 | 0.293 | 0.886 | 1.018 | 0.263 | 0.337 | 0.247 | 0.283 |
> > > > | FEDformer | 0.314 | 0.329  | 0.359 | 0.380 | 1.095 | 1.021 | 0.271 | 0.318 | 0.302 | 0.311 |
> > > > | Autoformer | 0.329 | 0.370 | 0.371 | 0.382 | 1.182 | 1.038 | 0.293 | 0.336 | 0.313 | 0.324 |
> > > >
> > > >
> > > > As before, NHiTS and NBEATS are strong competitors with PatchTST and significantly better than Autoformer. As there is not an enormous gap between the performance of PatchTST, TimesNet, NBEATS, and NHiTS, we do not think the conclusions of our paper would change upon including them in the analysis. We therefore see our baselining on DARTS and our additional experiments on the Informer dataset (in the rebuttal PDF) as a sufficient demonstration of LLMs' strong performance.
> > > >
> > > >
> > > > [1] Wu, Haixu, et al. "Timesnet: Temporal 2d-variation modeling for general time series analysis." arXiv preprint arXiv:2210.02186 (2022).
> > > >
> > > > [2] Challu, Cristian, et al. "NHITS: Neural Hierarchical Interpolation for Time Series Forecasting." Proceedings of the AAAI Conference on Artificial Intelligence. Vol. 37. No. 6. 2023.
> > > >
> > > > [3] Nie, Yuqi, et al. "A time series is worth 64 words: Long-term forecasting with transformers." arXiv preprint arXiv:2211.14730 (2022).

---

### Official Review · Reviewer_AFnc · 2023-07-05

**Soundness:** 2 fair
**Presentation:** 3 good
**Contribution:** 2 fair
**Rating:** 2
**Confidence:** 4

**Summary:**

This paper investigates to which extent an off-the-shelf large language model like GPT-3 can be used for time series forecasting. Its main claim is that forecasting performance can be very good, provided some care is taken in correctly encoding the input. Some limited experiments are proposed to support the claim.

**Strengths:**

- The paper is definitely timely in the sense that  large language models (LLM) are very trendy and that many other works already showed that they are good at transfer learning. In this respect, attempting to use them for forecasting makes perfect sense and this paper could possibly have a good impact.
- It cannot be denied that the reported performance is good, which supports the claim in a fascinating way.

There are some other limited contributions, notably regarding how the input should be encoded before using the LLM for forecasting, but I see them as very secondary when compared to the two mentioned above. Although I am recommending rejection for the reasons that I will describe below, I still must acknowledge that this paper asks a relevant and interesting question and is bound to meet some good impact, even if it is not presented at Neurips.

**Weaknesses:**

- If I had to summarize what I feel is my biggest concern with this paper, I would say that I feel uncomfortable with its _attitude_. As I mentioned above, it asks a question that I definitely think is interesting, but as it currently reads, it basically claims forecasting is solved by the method proposed, that outperforms state of the art in time-series forecasting.
In my view, the authors should have rather mentioned that using a LLM for forecasting interestingly seems to bring some very good performance in the experiments they did, but should have insisted much more on how much this should be understood today as some kind of funny curiosity at this stage. Indeed, there is something that is completely forgotten here: those LLM just cannot reasonably be used for forecasting in any realistic setup, that requires handling hundred thousands or million samples. What the paper convincingly shows is only that LLM can yield very fascinating performance for forecasting in the toy setups that were considered, and this is already a big deal.

- the paper contains many arguable statements that I will detail below. As a researcher in time-series, I must say that many statements there read pretty pejorative towards research in time-series and sound like "look how NLP will solve this problem that you have been struggling with for years".

**Questions:**

- L19 time series has
- L19 "time series are relatively unstructured": this is arguable. There is host of research precisely leveraging structure in time-series. For a very clear example, just consider speech, audio or music processing, where structure is everywhere.
- L30 "pretraining has not impacted time series forecasting to the same degree": pretty arguable and gratuitous statement
- L35: in a natural way
- L52 "instead of building towards a foundation model paradigm". I would say that this statement is not only arguable, it is very wrong and can even be felt as pretty pejorative for the host of researchers working in the field. The authors could for instance check out the many foundational books or papers that were written on modelling stochastic processes or in developing information theory and that all emerged in a time-series setup. Regarding recent foundational approaches, one could for instance mention neural processes or related fundamental research.
- L54-64: "uncertainty quantification for time series also tends to be domain-specific and ad hoc": I wonder what makes the authors think that NLP is free of any such bias. The whole paragraph has a strange pejorative feel to it that I cannot understand and that I think is inappropriate.
- L64: limited expressive
-  L65 "by directly numbers as strings of digits" ?
- Section 3.1: there is no reference whatsoever to source coding or arithmetic coding and their host of deep-learning based recent variants, which appear to me as a clear _foundational_ setup for the _adhoc_ method proposed here.
- L130-L139: Evaluation. It doesn't really make sense to me to compare the proposed RNN-method for quantization with parametric distribution models like GMM or Laplace, etc. The authors should have compared instead with a quantile-based model where the number of quantiles corresponds to the number of parameters of the RNN they fitted.
- The "occam razor" discussion from section 3.2 is interesting, where the authors show that in some toyish cases, a LLM is able to extrapolate a numerical sequence. Still, they then generalize this interesting feature they observed on toyish examples as a general feature of LLM, which is not supported in any way but is only a "working assumption" and should be mentioned as such.
- L149: "time series generated according to a deterministic function can be considered as only a small generalization of these numerical sequences": This is of course not true and the authors should check the host of different ways that were proposed in the past to model deterministic signals and their incredible complexity.
- L157: "maps on precisely to extrapolating linear trends" ?
- L168: it is important that the authors detail how the method is extended to autoregressively predict digit sequences to multiple numbers, since this is a very important ingredient of the approach.
- Figure 7: I do not understand where history length appears here
- Figure 8: As a transformer, I would expect TCN to actually easily handle missing values.

- In the conclusion: I would not say that the toy experiments proposed are enough to write that "LLM can act as extremely capable time series forecasters" in a _general_ manner. Once again, the paper makes a very nice case for very small sequences and asks a relevant question, but this remains only a funny curiosity for practitioners at this stage, considering how hard scaling to realistic time series appears. For instance, one second of audio sampled at 44.1kHz means 44100 samples. If each one ends up as a word, this would mean something like 50 pages of text, and this is only for mono. People routinely process hours of multichannel audio. Of course, other time-series practitioners have similar or even much more challenging scales to handle.
- The authors are apparently unaware of the fact that a very fruitful research has long been done on multimodal signal processing in time-series contexts.

**Limitations:**

What I wonder after reading this paper is whether time-series models could be useful for text processing.

--
After reading the rebuttal, I don’t think the authors can be trusted to make the huge amount of changes it would take to address my concerns.

---

> ### Author Rebuttal · Authors · 2023-08-10
>
> Thank you for taking the time to review our paper and for your detailed review. We appreciate your recognition of the paper's relevance, fascinating results, and potential for impact. We apologize if the tone of our paper comes off as dismissive of prior work. We never intended this to be the case. In several places we believe this perception may come about from a difference of terminology and perspective employed in the two research communities, for example the term “foundation models” which is commonly used to refer to large pretrained models [1], not to suggest that they are more foundational in nature than other research. We assure you that we respect the contributions and nuances of the time-series community and any perceived negative tone was unintended. While we acknowledge some of your concerns and will adjust the manuscript accordingly, including certain uses of language, there are other points we feel require further clarification.
>
> **Handling missing values with TCN:** We note that TCN stands for Temporal Convolutional Networks and is not a transformer. On the other hand, actual transformer based models may be able to more easily accommodate missingness, by using a designated token or omitting the missing entries. However, for trained transformer time series models, if missing entries are not encountered at training time, then with missing values the statistics inside the model will be different and the model will not be able to handle the missing entries gracefully. However, as our LLM-based method has not adapted to the statistics of fully observed time series (because it is only using in-context learning) it has a significant advantage in this scenario.
>
> **Uncertainty Quantification:** In our description of prior work in time-series uncertainty we agree that "ad hoc" was a poor choice of words (and commonly overused in machine learning papers). What we meant was that there is no standard approach that resolves the problem of parametrizing aleatoric uncertainty in time series models. Instead there are many different approaches, each with its own set of shortcomings, and no satisfactory singular resolution of this problem. We will revise this sentence on line 54 to express this viewpoint more clearly.
>
> **Time Series Structure:** Our remarks on time series being relatively unstructured primarily pertain to popular domains of applications like finance and climate. While these series have inherent structure, they are typically more stochastic in nature than data types like images, where deep learning has seen much more substantial success. Our aim wasn't to downplay the complexity of time series modeling. On the contrary, the lack of easily identifiable structures beyond basic ones like periodicity can itself be a challenge when developing forecasting models. We will ensure that our revised manuscript provides a clearer context to avoid any misinterpretations.
>
> **Pre-training in Time Series:** While pre-training has its applications in the time-series domain, its prevalence in NLP or vision is evidently much more pronounced. We don’t think it is fair to call our statement “arguable and gratuitous” without any specific evidence or references.
>
> **LLM’s Low-complexity Bias:** Our demonstration of LLM’s low-complexity bias in simple deterministic settings was to provide insight into why LLMs can perform well on real world timeseries which also contain low-complexity deterministic patterns, rather than to establish it as a universal phenomenon. Nevertheless, recent work such as [17] also supports our finding that LLMs exhibit a preference for simple completions with low Kolmogorov complexity.
>
> **Significance of Our findings:** We strongly disagree that our LLM-based time series prediction method is “only a funny curiosity”. While it’s true that it is not suitable for time series with a large number of samples, there are plenty of real prediction problems with short lengths as demonstrated by the DARTS and Monash datasets, on which LLMs perform among the best models.
>
> **Figure 7 History Length:** In the univariate setting, N-HITS and N-BEATS can learn a model by chunking up the time series into segments and training on the batched segments. When the history is short, not many chunks are available and it is insufficient for effectively training a neural network. Smaller models with less learnable parameters (e.g. ARIMA) tend to do better in this scenario, and LLMs also do very well because no parameters are being trained on the data (the inference is zero shot).
>
> **L168:** For the autoregressive prediction on the sequence, the tokenized numbers are just concatenated with comma separators as presented in section 4.  To compute the continuous NLL of the time series the total NLL is just the sum of the NLLs from each conditional distribution over a given number. Mathematically, $\mathrm{ContinuousLogLikelihood} = \log p(x_{1:T}) = \sum_{i=1}^T \big(\log p(x_i\in U_k|x_{1:i-1}) + nlog B\big) = \mathrm{DiscreteLogLikelihood}+ Tn\log B$ where $U_k$ is bin $k$.
>
>
> [1] Bommasani, Rishi, et al. "On the opportunities and risks of foundation models." arXiv preprint arXiv:2108.07258 (2021).

---

### Official Review · Reviewer_z6DB · 2023-07-06

**Soundness:** 3 good
**Presentation:** 3 good
**Contribution:** 3 good
**Rating:** 7
**Confidence:** 4

**Summary:**

This paper proposes a method to use LLMs for time series prediction.
Precisely, they develop a way to tokenize and encode numerical digits and prompt LLMs to generate future numbers given the past.
Their method is comparable to or exceeds the state-of-the-art time series models.

**Strengths:**

The paper proposes a very novel method to use LLMs for time series prediction. The problem is a high-impact problem, and their way of using LLMs is creative. I was excited when reading this paper.

The method is very carefully designed and is very convincing. Each component of the method (e.g., digit representation, tokenization) has been carefully thought through and demonstrated with insightful experiments and analysis.

The results are positive.

**Weaknesses:**

I would like to recommend this paper for acceptance, and I gave a positive score.

The main reason that I didn't give a higher score is its presentation and the limitation of the proposed method.

For presentation, I think the current structure needs improvement and some terminologies need clarification.

The current presentation is a little bottom-up: 3.1 explains how to represent numbers and 3.2 discusses how LLMs process and generate sequences of numbers; then 4 explains how these things are put together to form the method.
Honestly, I was lost until I read the end of page-5 (of a 9-page paper!) since I did not know why I had to read section 3 and I didn't know how 3.1 and 3.2 are connected.
Maybe a better way is top-down: give a high-level overview of the method first and then dive into each phase/component for their details. If the authors do not want to change the structure dramatically, maybe at least add a paragraph explaining the overall picture of the method at the beginning.

I would like the authors to give a precise clarification whenever they mention a term that has multiple meanings.
Maybe I am wrong but "multimodality" can mean "the distribution has multiple modes" or "the data has multiple forms". In this paper, it seems that the authors have used it for both cases (3.1 and 5.5) but that made me confused.

For limitation of the method, it seems that the proposed method requires something that modern blackbox LLMs do not like to offer: i.e., tokenization and log-likelihood computation. This reliance will restrict the impact of this work: this is fine, since I think the method is already very good and it is a reasonable limitation. But I would appreciate it if the paper has an aggregated section/paragraph discussing this limitation explicitly, instead of mentioning it here and there scattered across the paper.

**Questions:**

Please see my questions and suggestions in the Weakness.

**Limitations:**

Please see limitations in the Weakness.

---

> ### Author Rebuttal · Authors · 2023-08-10
>
> Thank you for your thoughtful and supportive review. Your comments about improving the readability of the paper through section 3 are well received. We have added a paragraph at the start of section 3 giving a high-level overview of the method, and we have revised the experiments in Figure 2 to use the decimal encoding. Hopefully these changes will make it clearer how the findings in 3.1 and 3.2 connect to the method we propose.
>
> On multimodality, your interpretation is correct and we have somewhat overloaded the usage of multimodal. We will retitle section 5.5 “Connecting Time Series and Language Understanding” to avoid confusion. Regarding the limitations of using black-box LLMs, we have added a complete limitations section in the general response discussing this and other issues that we will add to the main text of the paper. As you point out, tokenization and log-likelihood are often not accessible for the models exposed only through APIs (for example GPT-3 provides log-likelihoods but GPT-4 does not), and this can be a hindrance to the method and make comparison more difficult. However, the proliferation of highly performant open source models like LLaMA-2 provide reason for optimism on this front.
>
> Given that your main concern, and the reason for your weak accept rating, was the limitations of black-box LLM APIs, we hope you will considering raising your score in light of our new LLaMA results, which do not suffer from the same limitations.

---

> > ### Comment · Reviewer_z6DB · 2023-08-14
> > **acknowledge of rebuttal**
> >
> > Thank you for your response and new results. And thank you for considering my presentation suggestions.
> >
> > I would like to raise my rating to 7, with my trust that you will improve the presentation as you promised.

---

### Official Review · Reviewer_Zuxw · 2023-07-13

**Soundness:** 2 fair
**Presentation:** 2 fair
**Contribution:** 1 poor
**Rating:** 3
**Confidence:** 4

**Summary:**

This paper studies the usage of large language models such as GPT-3 for zero-shot time series forecasting task. To construct the LLMs input sequence in line with how they are pre-trained, it proposes to encode the continuous time series as a string of numerical digits with spaces added between single digits. As a result, time-series forecasting can be transformed into a next-token prediction task suitable for LLMs. Results on standard univariate time-series datasets show that LLMs can perform comparably to time-series models such as ARIMA, etc. in a zero-shot fashion i.e., without any finetuning.

**Strengths:**

1. The paper is well-written and most of the sections are easy to comprehend.

2. The studied problem is straightforward and interesting considering the capabilities exhibited by large language models.


**Weaknesses:**

1. Comparison to other recent transformer-based time-series forecasting methods (listed in the references below, but not limited to) is missing.

2. The paper considers simple univariate time-series forecasting datasets, it would be helpful to consider more complex multivariate time-series datasets and how they should be encoded.

3. In Appendix C, details about how other existing time-series forecasting models/methods perform compared to GPT-3 are missing on the time-series data that was recorded after GPT-3's training data cutoff date.

References:

[1] Informer: Beyond efficient transformer for long sequence time-series forecasting, Zhou et al., 2021

[2] Autoformer: Decomposition Transformers with Auto-Correlation for Long-Term Series Forecasting, Wu et al., 2021

[3] FEDformer: Frequency Enhanced Decomposed Transformer for Long-term Series Forecasting, Zhou et al., 2022



**Questions:**

Minor:

1. On page 4, I believe the figure references should be attributed to Figure 2 instead of Figure 3.


**Limitations:**

I could not find a section addressing the limitations and negative societal impact of this work.

---

> ### Author Rebuttal · Authors · 2023-08-10
>
> We thank you for your review and comments on the paper.
>
> **Transformer based time series methods** As we mentioned in the general response, we have added an additional evaluation of our model using the LLAMA-2 LLM which includes comparisons against Informer, Autoformer, and FEDformer as you suggested. We find that our LLM based time series prediction method compares favorably, with aggregate MAE values on par with or better than these transformer methods, but with completely zero-shot predictions.
>
> **Complex multivariate time-series** We have added a limitations section to the paper and we list this section in the general response. Our paper is focused on univariate time series, and as we discuss in the limitations section, multivariate time series are challenging to accommodate given the short context windows afforded by the LLMs. In the additional experiments on Autoformer benchmark datasets, we demonstrate that the LLaMA-2 LLM can perform well on multivariate data by considering it as a collection of univariate series. Developing methodology to accommodate all non-temporal correlations from the multivariate time series with LLMs is also a worthwhile endeavor, but we leave it to future work.
>
> **GPT-3 training cutoff date evaluation** We would like to direct you to figure 9 in appendix C where we evaluate the LLM predictions on three time series datasets recorded after the GPT-3 training data cutoff date. We evaluate the predictions both qualitatively (showing forecasts and prediction intervals) and quantitatively (using the negative log likelihood). Measured by NLL, GPT-3 performs strongly on the three datasets, and the forecasts are consistent with performance on the other time series we evaluate in the paper.
>
> **Limitations** We have added a limitations section in the general response that we will be adding to the main text.
>
> As you mention, we believe the question of performing time series predictions with LLMs is of significant interest. We hope that given the new experiments performed and clarification of your concerns you would consider raising your score.

---

### Official Review · Reviewer_s41P · 2023-07-14

**Soundness:** 3 good
**Presentation:** 2 fair
**Contribution:** 3 good
**Rating:** 5
**Confidence:** 3

**Summary:**

The authors propose a way to use autoregressive large language models to solve time series problems. Several tricks make the time series inputs amenable to LLMs (e.g., separating digits by a space to circumvent tokenization problems). GPT-3 outperforms existing methods on several time series benchmarks. LLMs can also perform imputation on missing data in the time series.

**Strengths:**

The paper considers the interesting setting of applying pre-trained LLMs to "classical" (i.e., 1-D) time series problems. The tricks introduced to format the time series as input to an LLM are interesting in their own right, though I am not fully sure if they are novel or not. The authors also conduct thorough ablations on these tricks to see how they might affect the performance of GPT-3 on these types of problems.

**Weaknesses:**

**Costs of using LLMs**

One crucial discussion missing from this paper is the inference cost. If using GPT-3 or GPT-4, then this is a literal payment. Even for open-sourced LLMs, I would imagine that the inference cost is much higher than that of other time series models. Similarly, using LLMs for imputation is much more expensive than standard methods (e.g., interpolation). The authors should acknowledge this weakness clearly.

**No comparison to transformer time series models**

The authors ought to include comparisons to other transformer time series models, such as the Informer and Autoformer, which were clear leaps forward in the field. It's hard to tell if the gains observed in this paper are from the massive amounts of pre-training data (as suggested by Section 3.2?) or from the self-attention mechanism, which does seem well-adapted to time series. Please also compare the size of GPT-3 to those models. This is my main reason to not give it an accept rating.

**Writing**: Please define MAE, CRPS, etc before using them in the text. The colors used for "Ground Truth" and "GPT-3" are too close to each other in Figure 3, so it is hard to read. The authors should include a discussion of how positional embeddings make it easier for an LLM to interpret the input data and operate across the time domain.

**Questions:**

**I don't understand the argument in Section 3.2: LLMs extrapolate patterns in numerical sequences.**

The discussion in Section 3.2 is confusing to me. The authors suggest that LLMs can handle sequences with linear growth, exponential growth, etc. What's confusing is that these are all fixed functions that have no dependence on the previously occurring values. Something that may be more interesting is e.g. the Fibonacci sequence, where the next value is a function of past values. I don't really get the point of Figure 3 + Table 1 because these are just simple functions, and if I wanted to fit them, I would directly fit them using the appropriate function class (i.e., not huge, overparametrized neural networks).

I also don't really get why it is a given that LLMs can do "simple arithmetic operations". Although these are simple to us, they may be complicated functions to learn in the embedding space, and they depend on the tokenization used. Note that the cited paper [17] is  focused on low-complexity sequences in the sense of Kolmogorov complexity, which doesn't necessarily correspond to simple arithmetic operations.

**Can the authors motivate their dataset choices?**

The transformer-related time series models don't use the DARTS dataset.

**Limitations:**

**Not enough LLMs**

The writing indicates that the authors are trying to make the broader claim that all LLMs (that can perform in-context learning) can serve as time series models. To actually substantiate this claim, they need to test other LLMs besides GPT-3 and GPT-4.

---

> ### Author Rebuttal · Authors · 2023-08-10
>
> We thank you for your thoughtful and detailed review. In our understanding, your main concerns about the paper are the need for testing additional LLMs, the lack of comparison against Informer and Autoformer, and the purpose of the argument that we present in section 3.2.
>
> **Evaluating other LLMs** In fact we did evaluate other available LLMs shown in Figure 7 (right) and in Appendix B.3, though to further address your question, we have added the recent LLaMA-2 70B model to our investigations and find that in many cases it outperforms GPT-3 Davinci. These comparisons are shown in Figure 1, 2, and 3 of the attached PDF.
>
> Regarding your point about transformer methods like Informer and Autoformer, we have added comparisons against these methods shown in Figure 1 in the attached PDF. We compare LLaMA-2 70B with Autoformer, Informer, Reformer, and a transformer model on 5 of the autoformer benchmark datasets with two different prediction lengths. We find that LLaMA-2 70B performs on-par with Autoformer and better than Informer in aggregate. We were surprised to find that the model performed this well out of the box without any hyperparameter tuning or modification of the sampling temperature. While we think this performance is quite impressive, it’s worth noting that the LLaMA model is about 10,000 times larger than the autoformer models it is compared against, and the LLaMA model appears to experience degrading performance with longer sequence lengths, but the LLaMA model is predicting completely zero-shot and is capable of performing all kinds of text-generation tasks in addition to forecasting.
>
> **Inference costs** Regarding inference costs, because we do not perform any training of the base LLM, these are actually quite manageable. A single forecast with maximum history length costs only 4 cents with the largest GPT-3 Davinci model. In general these costs are very low, except if one requires to draw many many samples from the distribution over trajectories. In total we have spent less than 1000 dollars on GPT-3 queries for prototyping our method and evaluating on the many datasets.  With the ability to run models locally like LLaMA-2, this cost decreases even further.
>
> **Section 3.2** Regarding Section 3.2 and the deterministic synthetic time series, these experiments provide a controlled environment for uncovering which patterns are easier for the LLMs to identify and extrapolate, and provide an understanding of why this is the case. Ultimately we do not care about the model’s performance on these deterministic signals for their own sake, and of course one could fit these functions very well by directly parameterizing a function in the right family. Although these functions are easy to express as a mathematical formula, as you allude to, that does not mean that the functions are easy to express at a token level inside an LLM and even less so when the task is combined with identifying the recurring pattern from the history. Hence, the ability to identify and extrapolate these signals is a nontrivial capability of the language models and we try to make sense of how this is possible in simple cases through the arithmetic operations that are entailed. While the discussion is focused on Kolmogorov complexity in [17], the experiments on the low complexity bias of LLMs actually measure complexity with respect to a reduced language of expressions that can be formed from an expression tree with constant, addition, multiplication, and integer division primitives. These primitives coincide with those we discuss in Section 3.2 for simple pattern extrapolation.
>
> Given that we have performed the necessary experiments to specifically address your concerns, we hope that you will consider raising your score.

---

> > ### Comment · Reviewer_s41P · 2023-08-11
> >
> > Thanks to the authors for the additional experiments as well as the limitations section in the general response. My original concerns about inference cost are proven to be true, and the limited context window of LLMs does indeed make them impractical as a practical method.
> >
> > I took some time to read through the other reviewers' comments and realized I was not familiar with PromptCast (mentioned by Reviewer URMj). After reading that paper, I feel that the interesting insights from this paper have already been shown in that work. Namely, I think the main contribution of this work is to show that LLMs can perform time-series forecasting. The authors do not cite this work and instead imply theirs is the first to consider using LLMs for this setting. PromptCast was posted >6 months before submission, so I am curious to hear why the authors did not discuss it. I found their response to Reviewer URMj to be unsatisfactory, because it primarily focused on the idea that this paper advances some understanding of why LLMs can do time series forecasting, which I take issue with (see below).
> >
> > My concern with Section 3.2 remains. I don't think that studying LLM behavior on these highly simple primitives is indicative of their behavior on more complex data (like the Fibonacci sequence I suggested). Section 3.2 gave me little to no additional understanding of why LLMs can operate on time series. If the authors would like to make this point, I suggest they study more complex settings.
> >
> > Given the concerns with PromptCast and Section 3.2, I won't be raising my score on this paper.

---

> > > ### Author Response · Authors · 2023-08-13
> > > **Addressing PromptCast and Section 3.2 (part 1)**
> > >
> > > **PromptCast**
> > >
> > > We did not discuss PromptCast in our original submission because we were not aware of it at the time. From a methodological standpoint, it is worth noting that our approach to forecasting with LLMs is very different from the method introduced in PromptCast. Let’s use the series of numbers [1,2,3] as example inputs to the model and [4,5] as example outputs. PromptCast takes raw values and inserts them into a text prompt that is specifically designed for a given dataset. For example, the following input and output prompts (taken verbatim from the paper) are used for the “City Temperature” dataset:
> > >
> > >     Input: “The average temperature of was 1, 2, 3 degree on each day. What is the temperature going to be on tomorrow?”
> > >     Output: “The temperature will be 4 degree.”
> > >
> > > Using our method with precision 3 would instead have the following inputs and outputs:
> > >
> > >     Input: “0 , 5 0, 1 0 0 ”
> > >     Output: “1 5 0 ”
> > > ​​
> > > As you can see, the two methods are quite distinct. We used the very specific formatting in our paper because it led to the best results. We did not find additional prompting necessary or helpful, and we found that rescaling and paying careful attention to tokenization was absolutely essential to good performance. Our method also has the additional benefit of being directly applicable to any input dataset without dataset-specific prompt engineering, which is evidently necessary in the PromptCast framework. Our method is also more robust to the scale of the input numbers and doesn’t require any knowledge of their associated units.
> > >
> > > Beyond clear methodological differences, the approaches to evaluation are very distinct between our work and PromptCast’s. Firstly, their metrics are purely deterministic and therefore do not effectively capture whether language models have learned a proper distribution over the continuous values. Deterministic metrics are known to be problematic in time series analysis, because incredibly naive forecasts can often perform well only considering point estimates [1,2]. For example, simply predicting the last value in the input or the mean of the input values can perform better than many “state-of-the-art” methods under deterministic predictions. For this reason, we spent a significant fraction of our submission assessing the ability of language models to fit the continuous time series distributions. This analysis, and the associated strong performance of LLMs on distributional metrics like negative log likelihood (NLL) and continuous ranked probability score (CRPS) is unique to our work and is absent from PromptCast.
> > >
> > > In addition to the lack of probabilistic evaluation, PromptCast also provides much weaker evidence of generalization across datasets–only examining 3 datasets, while we evaluate our method on well over 20 unique datasets, including standard benchmark datasets like DARTS, Monash, and the Informer dataset. We also anticipated concerns about data leakage (potential memorization of test data by the LLM) and provided additional experiments demonstrating strong LLM performance on datasets collected after the training cutoff, while PromptCast performs no such analysis and therefore offers no reassurance that its result are not simply a side-effect of data leakage.
> > >
> > > It’s notable that URMj was the reviewer to mention PromptCast as an alternative method and still gave our submission a positive rating. Despite surface-level similarities, we think there are fundamental differences between PromptCast and our submission and that the unique contributions of our submission make it worthy of acceptance.

---

> > > > ### Author Response · Authors · 2023-08-13
> > > > **Addressing PromptCast and Section 3.2 (part 2)**
> > > >
> > > > **Section 3.2**
> > > >
> > > > We can quickly verify that LLMs do indeed generalize to more complex sequences, including the Fibonnaci sequence, Tribonacci sequence, or Lucas sequence. In the examples below, we take 20 samples from GPT-3 with our method, rank them with likelihood, and then show the highest likelihood sample:
> > > >
> > > > Fibonacci sequence:
> > > >
> > > >     Input: [3, 5, 8, 13, 21, 34]
> > > >     Ground truth: [55, 89, 144, 233, 377, 610, 987, 1597, 2584, 4181, 6765, 10946, 17711, 28657, 46368, 75025, 121393, 196418]
> > > >     Sample: [55, 89, 143, 243, 381, 618, 1027, 1647, 2558, 4196, 6655, 10656, 17069, 27174, 44283, 71454, 115968, 188377]
> > > >
> > > > Lucas sequence:
> > > >
> > > >     Input: [1, 3, 4, 7, 11, 18, 29]
> > > >     Ground truth: [47, 76, 123, 199, 322, 521, 843, 1364, 2207, 3571, 5778, 9349, 15127, 24476, 39603, 64079, 103682, 167761]
> > > >     Sample: [46, 79, 132, 213, 344, 557, 901, 1468, 2357, 3825, 6186, 9803, 15789, 25592, 41481, 66780, 107479, 176268]
> > > >
> > > > Tribonacci sequence:
> > > >
> > > >     Input: [1, 1, 2, 4, 7, 13]
> > > >     Ground truth: [24, 44, 81, 149, 274, 504, 927, 1705, 3136, 5768, 10609, 19513, 35890, 66012, 121415, 223317, 410744, 755476]
> > > >     Sample: [24, 41, 73, 131, 243, 416, 736, 1334, 2446, 4173, 7390, 13398, 24485, 41836, 74579, 149868, 266299, 418772]
> > > >
> > > > As we can see, GPT-3 provides a reasonable approximation of these sequences, as it did with an exponential function. The approximation deteriorates for large numbers most likely because current LLMs have trouble implementing addition and multiplication between large integers in a zero shot manner (though fine-tuning can increase these capabilities significantly [3]). We also think it’s worth noting that examples like exponential and sigmoid functions are directly relevant to common phenomena that underlie time series data, such as the spread of viral diseases, while the above sequences are less directly relevant, and can actually be expressed as exponentials themselves (matrix exponentials).
> > > >
> > > > **Connection with complexity**
> > > >
> > > > The important insight of papers like [4] is that large language models learn a useful distribution over likely sequences during pretraining. This distribution assigns higher likelihood to completions that follow from the input via a simple rule and lower likelihood to more complex rules. Therefore, when we prompt an LLM with values that obey the fibonacci sequence, it is much more likely to sample completions that also generated by a simple rule. If we sample the LLM sufficiently, we will also sample completions that follow a different pattern, but they are far far less common. In time series forecasting, the same principle applies. Forecasts are more plausible if they can be explained by extending simple patterns in the data (ones that are less likely to be artifacts of noise or limited observations).
> > > >
> > > > [1] Hansika Hewamalage, Klaus Ackermann, and Christoph Bergmeir. Forecast evaluation scientists: common pitfalls and best practices. Data Mining and Knowledge Discovery, 37(2): 788–832, 2023.
> > > >
> > > > [2] Yue Cui, Jiandong Xie, and Kai Zheng. Historical inertia: A neglected but powerful baseline for long sequence time-series forecasting. In Proceedings of the 30th ACM International Conference on Information & Knowledge Management, pages 2965–2969, 2021.
> > > >
> > > > [3] Liu, T., & Low, B. K. H. (2023). Goat: Fine-tuned LLaMA Outperforms GPT-4 on Arithmetic Tasks. arXiv preprint arXiv:2305.14201.
> > > >
> > > > [4] Goldblum, Micah, Marc Finzi, Keefer Rowan, and Andrew Gordon Wilson. "The No Free Lunch Theorem, Kolmogorov Complexity, and the Role of Inductive Biases in Machine Learning." arXiv preprint arXiv:2304.05366 (2023).

---

### Official Review · Reviewer_URMj · 2023-07-21

**Soundness:** 3 good
**Presentation:** 3 good
**Contribution:** 2 fair
**Rating:** 6
**Confidence:** 2

**Summary:**

The papers presents an analysis of larger language models for the task of time series prediction. The paper claims that these large language models being very generalizable are able to work very efficiently with numbers in these tasks since each digit prediction is a conditional over the previously predicted digits and hence acts in a way similar to hierarchical softmax to predict the final output.



**Strengths:**

Even though a lot of the work presented here is known to be useful, less is known about how these models work and performing a closer analysis with simplified datasets and well defined examples gives a much better understanding about what works and what does not and helps figure out areas of improvement.

For example the approach to split numbers by digits, previous works have achieved similar results by removing multi-digit entries from the vocabulary of the model which eventually has the same impact as this paper. But this paper presents intuition around how this might help the model in predicting one digit at a time and be more performant using the analogy to heirarchical softmax.

The well defined set of experiments show that this approach works very well to improve the ability of LLMs to work with numbers and the analysis on time series datasets confirms the same.

**Weaknesses:**

There are already papers showing that:
1. Mathematical ability of LLMs improves significantly if we process digit by digit (approached generally by removing multi-digit entries from the vocab).
2. Application of LLMs on time series datasets which generalize better then regular approaches has also been shown.

See "PROMPTCAST: A NEW PROMPT-BASED LEARNING PARADIGM FOR TIME SERIES FORECASTING"

So there isn't much novelty provided by this paper.

**Questions:**

Some typos
- line 59: "from from" -> "from"
- line 63-64: "without any major changes by directly numbers as strings of digits." --> needs to be reworded

**Limitations:**

- Since we depend on the LM model to perform the forecasting, it is much harder to understand why a particular forecasting was chosen. There are always multiple ways to interpret a series for completion and relying that the LM with follow Occum razor is dangerous. Only the other hand, traditional time series forecasting methods are much more interpretable.

---

> ### Author Rebuttal · Authors · 2023-08-10
>
> Thank you for the supportive review and your comments on related work.
>
> **Comparison to related work**
>
> Though other papers have investigated the use of LLMs in basic arithmetic (e.g Goat [1]) and forecasting (e.g. PromptCast), we believe the findings of our paper are unique in their scope and focus. We delve deep into understanding why LLMs can extrapolate on time series, emphasizing their bias for low-complexity completions, ability to identify and execute arithmetic operations on numerical sequences, and superior representation of uncertainties. We also uncover scaling results and the effect of alignment methods which are not present in related work (e.g. PromptCast).
>
> **Following Occam’s Prior**
>
> Regarding your comment on Occam’s razor, of course there are many consistent completions of a time series (just as there can be many consistent extrapolations of a numerical pattern). Following the Occam’s razor prior explored in [17] does not mean that we should _select_ the simplest trajectory consistent with the observed history, but rather that we should form a predictive distribution and _sample_ according to their posterior likelihoods using this prior conditioned on the observations, resulting in simpler completions being sampled more often. This process is formalized through Solomonoff induction, whereby the Occam’s razor prior (also known as Solomonoff prior) $p(x) = 2^{-K(x)}/Z$ (where $x$ is the complete time series and $K$ is the prefix Kolmogorov complexity) is used as a prior to form a posterior $p(x|x_{[:t]}) = p(x_{[:t]}|x)p(x)/p(x_{[:t]}) \propto 2^{-K(x)}$ for all series $x$ that match the history. We believe that LLMs capture a very rough and approximate version of this Solomonoff prior. With the deterministic time series in Figure 3, these signals can be explained by a generating program of short length, and other programs which correctly fit the previous history but make substantially different predictions on the future require many more bits to express. These solutions are still represented in $p(x|x_{[:t]})$ but are just exponentially less likely to get sampled (if we sample enough we should recover them). On the other hand, for real time series that have noise, the shortest generating program for the data is still very large since it must encode the noise. Therefore, other explanations of the data are close in size and therefore sampled frequently in the posterior distribution of the Occam’s prior, and that reflects what we see in the LLM predictions in e.g. Figure 6 that have considerably more variation.
>
> [1] Liu, T., & Low, B. K. H. (2023). Goat: Fine-tuned LLaMA Outperforms GPT-4 on Arithmetic Tasks. arXiv preprint arXiv:2305.14201.

---

> > ### Comment · Reviewer_URMj · 2023-08-16
> >
> > Thanks for your clarifications!
> >
> > The paper is structured in a way that it focuses on presenting LLMs as a viable solution for timeseries forecasting, and not what is mentioned above. The overlap with PromptCast is significant and IMO needs to be better acknowledged in the paper.
> >
> > In #2, I wanted to focus on the fact that when we compare the advantages and disadvantages of traditional and LLM based approach for time series forecasting, I agree that LLMs are better in many aspects, but we also need to point out that interpretability of the outputs generated through traditional methods is a major advantage for these methods where LLMs lack significantly.

---

> > > ### Author Response · Authors · 2023-08-21
> > > **Follow-up notes**
> > >
> > > Thank you for your engagement and thoughtful suggestions throughout the review process.
> > >
> > > **Overlap with PromptCast**
> > >
> > > We have provided a more in-depth comparison between PromptCast and our method in our latest general comment (entitled “Our Submission’s Contribution”), which we hope you will read over and consider. We show that PromptCast performs much worse than our approach in practice, and we detail many important differences both at a high-level and in the methodological details.
> > >
> > > **Interpretability of LLMs vs traditional methods**
> > >
> > > We agree that interpretability is a major appeal of traditional methods with a small number of parameters (e.g. ARIMA). It’s worth noting that many popular deep methods for forecasting (e.g. DeepAR, TCN, PatchTST) already sacrifice this interpretability for predictive performance and yet have been widely adopted. Our question-answering experiments in Section 5.5 also suggest another mechanism for achieving interpretability with LLM predictions. We show that LLMs can be queried in English to recognize deterministic trends. It is therefore also possible that LLMs can be effectively prompted (or fine-tuned) to provide useful explanations of their predictions or deconstruct high-level trends.

---

### Official Review · Reviewer_eErN · 2023-07-23

**Soundness:** 3 good
**Presentation:** 3 good
**Contribution:** 4 excellent
**Rating:** 7
**Confidence:** 4

**Summary:**

Briefly summarize the paper and its contributions. This is not the place to critique the paper; the authors should generally agree with a well-written summary.
This paper proposes that LLMs like GPT-3 can be employed as time series forecasters due to the similarities between text and time series such as periodicity. The authors show how LLM can be used to model multimodal distributions through a hierarchical discretization of the output space and uniform distribution in each discrete bucket, enabling training of LLM with negative log-likelihood. The hierarchical discretization also allows the LLMs to circumvent the requirement of a fixed output range with binning values and a large number of bins for an expressive output space.
Experiments show that GPT-3 fitted with the proposed modifications can forecast time series (e.g. linear, periodic, exponential) better than baselines such as TCN and ARIMA. The authors also explore optimal ways to encode time series data in LLMs and found that using a base of 10 and use of space to individually tokenize digits in numbers result in better performance. When benchmarked against existing models, the proposed approach perform competitively with top baselines in the DARTS and Monash datasets. Further experiments were conducted to show that LLMs as time series forecasters are sample efficient, have performances scaled with model size and handles missing data well.


**Strengths:**

The approach to using LLM as a time series forecaster is innovation and the authors found ways to elegantly unlock this capability without the need for extensive effort (e.g. retraining or finetuning).
This finding will be interesting and relevant to a wide audience in this community.
Experiments were extensively conducted to show optimal approaches for LLMs’ time series forecasting ability and other aspects such as sample efficiency etc.


**Weaknesses:**

Limitations of the proposed approach were not discussed in the paper

**Questions:**

a) The experiments seem to be mostly in a zero-shot setting, have the authors studied the few-shot setting where few-shot examples are in the prompt and how would that affect forecasting performance?

b) What prompts, if any, are given to the LLM, other than the initial time series data and what effects would the choice of prompts have on its performance?


**Limitations:**

No, a separate section addressing the approach's limitations and potential societal impact would be recommended.

---

> ### Author Rebuttal · Authors · 2023-08-10
>
> Thank you for your supportive review!
>
> To answer your questions,
> We did not study the few-shot setting, but it would be a great direction for future work. We suspect that both few-shot prompting and fine tuning could substantially improve forecasting performance. One challenge of few-shot prompting, however, is the limited context window of current LLMs. In the zero-shot setting, we can use the full context window to extend the number of past observations used by the model. In the few-shot setting, this same context window might get consumed by the few-shot examples, limiting the number of past observations.
> All the results in paper that use GPT-3 or LLaMA models are obtained without any natural language prompting. We experimented with prompting with a description of the dataset, but the initial results showed no improvement in performance. For ChatGPT and GPT-4, we used the following prompt (which was necessary for good forecasts): "You are a helpful assistant that performs time series predictions. The user will provide a sequence and you will predict the remaining sequence."
>
> We appreciate the question about limitations. There are several key limitations that are worth noting and we have added them into a unified section shown in the general response, which will be integrated into the paper. The limitations of our method are largely the limitations of LLMs in general, specifically limited input sequence length (small context window), subsequent challenges in handling multivariate data, and computational cost. Additionally, black box APIs have limited control over system prompts, RLHF, tokenization and likelihoods. As each of these limitations is an active area of research in itself, we expect that all progress should also improve the practicality of our method. By connecting LLMs with time series forecasting, we have also opened new avenues in evaluating progress in LLM performance on a task that requires understanding complex, quantitative patterns.

---

> ### Comment · Reviewer_eErN · 2023-08-12
> **Acknowledgement of Rebuttal**
>
> I have read the authors' rebuttal and decided to keep my original score.

---

### Official Review · Reviewer_duBe · 2023-07-28

**Soundness:** 2 fair
**Presentation:** 2 fair
**Contribution:** 1 poor
**Rating:** 3
**Confidence:** 3

**Summary:**

This paper discusses a novel approach to time series forecasting using large language models (LLMs) like GPT-3. The authors encode time series as a string of numerical digits and view time series forecasting as next-token prediction in text. They find that LLMs can extrapolate time series at a level comparable or exceeding the performance of several classic time series models. The authors argue that LLMs excel at extrapolating deterministic numerical sequences because of biases for simple explanations introduced by their text pretraining.  The authors suggest that this approach has potential and enables new capabilities, such as integrating non-numerical text into the input or output.

Overall, the paper studies an interesting problem setting that directly utilizes the in-context learning ability of Transformers to conduct zero-shot forecasting problems. In particular, the pre-trained transformer model (e.g., LLMs) can give promising performance. However, the analysis in the paper is not comprehensive enough. First of all, the paper fails to answer why pre-trained LLM can yield better performance. The arguments in the paper either focus on RNN (e.g., L121) or directly assume LLMs can perform extrapolation on time series sequences (e.g., L144 and L154). Based on the current presentation, it looks like the authors design a specific feature engineering method (digits representation) that can properly utilize the LLMs' extrapolation or in-context learning ability. If it is the case, the most important research question, why pretrained LLMs can conduct arbitrary time series' extrapolation better than other models is not answered or well discussed. From my perspective, if the major innovation of the paper is a feature engineering method, the current numerical performance would not be strong enough to lead to a publication in the top machine learning conferences like Neurips.



Moreover, the experimental settings in the paper contain learnable parameters that are tuned based on the forecasting error. From my perspective, I would not treat it as zero-shot learning setting, since parameters can be viewed as trained. The authors also don't provide codes to help reviewers verify the results of experiments.

In summary, this paper studies an interesting problem that utilizing pretrained LLMs to help time series forecasting. However, based on the current presentation, the analysis and the numerical performance are not strong enough.

**Strengths:**

This paper studies an interesting problem that utilizing pretrained LLMs to help time series forecasting

**Weaknesses:**

Please see my comments in Summary section.

**Questions:**

Please see my comments in Summary section.

**Limitations:**

The authors don't directly discuss the limitations of the proposed methods in the main paper.

---

> ### Author Rebuttal · Authors · 2023-08-10
>
> We thank you for your review.
>
> **Understanding why LLMs can perform extrapolation on time series:**
> We have demonstrated empirically that LLMs indeed can perform extrapolation on both deterministic and stochastic time series in Section 3.1 and 3.2.  We explicitly pose the ability to extrapolate simple deterministic trends as the ability to identify and implement simple arithmetic operations on the numbers in the sequence, for example addition by a constant for linear growth, multiplication for exponential growth, or copying for periodicity. Prior works [4,5] have shown that larger LLMs are increasingly capable at performing addition and multiplication at the token level, and the repetition bias is also a well-studied phenomenon [3]. We also relate these well-known phenomena with recent findings that LLMs have a preference for numerical sequences generated by a small number of arithmetic operations [2]. While we devoted a significant portion of the paper to investigating why LLMs are well suited to time series prediction, ultimately a good theory will hinge on a deeper understanding of the inductive biases of the transformer architecture and the effects of extensive generative pretraining. These questions are among the most important open problems in the field of deep learning at large and extend beyond the scope of our paper.
>
> **Comprehensiveness of the analysis**
> RNN vs LLM for Figure 2 experiments (fitting continuous distributions): For figure 2 we chose to use an RNN as the autoregressive model over the digits in order to disentangle uncertainty representation from a model’s ability to perform in-context learning. While we can train a small RNN or transformer directly to fit the distribution via gradient descent, the practical setup required in time series requires adapting to the distribution purely using in context learning, a considerably more challenging task. In section 3.1 we wanted to evaluate this parametrization of continuous densities in isolation. We have previously run GPT-3 on this identical problem with in context learning and the results are similar (though slightly worse due to the burden of only using in context learning).
>
> **Zero shot setting:**
> In general we find that there is little value in tuning the hyperparameters of our LLM time series prediction method on the data at hand. Using a fixed set of hyperparameters reproduces very nearly the same performance, and in the additional LLaMA-2 experiments we always use the fixed sampling parameters temperature (T) = 1.0, nucleus size (top_p) = 0.9, and 3 decimals of precision. Regardless, by zero shot we simply mean the setting without additional data besides the given series, for example no other instances of completed or uncompleted time series, and we feel that this usage is consistent with how zero shot is used in computer vision and NLP ( e.g. a zero shot image classification method that ingests a test image and performs test-time fine-tuning [1]).
>
> **Code:**
> We fully intend to release the code for our method on the acceptance of the paper.
>
> **Limitations:**
> While we do discuss several limitations within the paper (such as the limited context window, and the challenge of multivariate series), we recognize that it would be better to have these limitations aggregated into a dedicated limitations section. We have listed this new limitations section in the general response, and we will add it into the revised version of the paper.
>
> [1] Shu, Manli, et al. "Test-time prompt tuning for zero-shot generalization in vision-language models." Advances in Neural Information Processing Systems 35 (2022): 14274-14289.
> [2] Goldblum, Micah, Marc Finzi, Keefer Rowan, and Andrew Gordon Wilson. "The No Free Lunch Theorem, Kolmogorov Complexity, and the Role of Inductive Biases in Machine Learning." arXiv preprint arXiv:2304.05366 (2023).
> [3] Holtzman, Ari, et al. "The curious case of neural text degeneration." arXiv preprint arXiv:1904.09751 (2019).
> [4] Yuan, Z., Yuan, H., Tan, C., Wang, W., & Huang, S. (2023). How well do Large Language Models perform in Arithmetic tasks?. arXiv preprint arXiv:2304.02015.
> [5] Liu, T., & Low, B. K. H. (2023). Goat: Fine-tuned LLaMA Outperforms GPT-4 on Arithmetic Tasks. arXiv preprint arXiv:2305.14201.

---

> > ### Comment · Reviewer_duBe · 2023-08-20
> > **Thanks for your comments.**
> >
> > I would like to express my gratitude to the authors for acknowledging and taking into account my concerns. After carefully evaluating the responses and considering the feedback provided by other reviewers, my rating remains unchanged for now. As the discrepancy between the authors and myself lies in a subjective aspect of novelty/contribution, I would like to leave it to the reviewer-reviewer/AC discussion in order to align my understanding with the review team.  A final decision will be made following further discussions with the other reviewers and the AC.

---

### Author Rebuttal · Authors · 2023-08-10

General Comment to All Reviewers

We are thankful for the thoughtful and supportive feedback. We were happy to see many of the reviewers think that the goals and results of the paper are interesting, relevant, and useful. Few ML researchers would confidently assert that language models are zero-shot time series forecasters. This finding is simultaneously surprising and supported by diverse experiments on real-world datasets. To unpack this result, we perform a comprehensive set of experiments to understand why language models are well-suited to forecasting and what factors impact the performance. Because there were a few recurring themes in reviewer questions, we would like to make a few general points before responding individually.

**Evaluation of Other Large Language Models** Several reviewers mentioned the need for evaluating other LLMs to better demonstrate the generality of the method. We would like to highlight Figure 7 (right) in the original submission’s main text, where we compare GPT-3 against many other LLMs that are accessible via open APIs, evaluated on the Darts datasets. Additionally, we are happy to share new results with the recent open-source LLaMA-2 models. These results, shown in Figure 1,2,3 of the attached PDF, demonstrate very strong performance, with metrics comparable to GPT-3 on Darts and significantly exceeding GPT-3 on Monash, making it the best performing model overall on those datasets. These results help demonstrate that our method is not limited to GPT-3, that more powerful LLMs will likely yield continued improvements on time series prediction, and that the limitations of working with LLMs locked behind APIs can be alleviated with open source models.

**Additional Comparisons Against Transformer Time Series Models** Following suggestions from the reviewers, we have added comparisons against the recent Autoformer, Informer, Reformer, and FEDFormer methods. LLaMA-2 demonstrates strong performance against these models across the board.

**Additional Limitations Section** Several reviewers suggested that it would be helpful to have a unified limitations section. We present the following limitations section below that we will add into the revised version of the paper:

“While we have demonstrated that text-pretrained LLMs are surprisingly capable at time series extrapolation, our approach has a few notable drawbacks, resulting from limitations of the underlying LLMs. Most notably the relatively short context window of LLMs limits the amount of information that can be processed. Our approach is therefore best suited for time series that are short in length, univariate, and consist of a single instance. Longer time series are currently truncated in order to fit into the context window (though subsampling could also be used), and multivariate time series pose an additional challenge, as the length is multiplied by the number of covariates. One possible solution for multivariate time series is to make predictions on the distinct dimensions separately, ignoring the correlations between the channels, but this independence assumption is less than ideal.

Computational cost can also be a limiting factor. While most time series models are lightweight and fast to run over a large number of series, LLMs are significantly more expensive, even when used solely for inference. Beyond the properties of LLMs themselves, relying on API access to black-box LLMs can lead to its own set of drawbacks, as we lack full control over tokenization and model alignment methods. As we show in the paper, correct tokenization is essential to good forecasting performance and safety-driven prompts and RLHF may directly interfere with forecasting ability. Black box models also increasingly do not provide likelihood values (GPT-3 does, GPT-4 does not) that are necessary for computing the continuous probability density functions for time series. While these are significant obstacles when using LLM APIs, we hope that open source LLMs such as LLaMA can alleviate many of these issues, enabling direct control over prompting, tokenization, and likelihood calculations.”

---

### Author Response · Authors · 2023-08-21
**Our Submission’s Contribution**

**Making our case for acceptance**

Our submission contains at least 4 unique and impactful contributions
1. A method for turning a pre-trained LLM into a forecaster and continuous density estimate using careful formatting.
2. A theory and experiments explaining the observed success of LLMs, based on their simplicity bias and uncertainty representation.
3. Evaluations with over 20 unique datasets and over 20 baseline methods, including both deterministic and probabilistic metrics, as well as an analysis of model scaling and human alignment methods (e.g. RLHF).
4. A demonstration of the unique, text-based abilities of LLMs, including  time series question-answering and seamlessly handling missing data.

**Relationship with prior work**

Several reviewers have asked us to comment on differences between our work and “PromptCast: A New Prompt-based Learning Paradigm for Time Series Forecasting” [1]. Though we were not aware of it at the time of submission, PromptCast is certainly related work and the similarities demand a more detailed comparison.

**Methodological differences**

Both PromptCast and our method use pre-trained language models to perform forecasting, but most of the practical details are different. Below we show an example of each method, which we use to explain the differences:

PromptCast

    Input: “The values in the WoolyDataset for the past 95 time steps are 6172, 6709, 6633, 6660, 6786, 6800, 6730, 6765, 6720, 7133, … , 6590, 5590, 5135, 5762, 6077. What will the values for the next 5 time steps be? The values for the next 5 time steps will be”
    Output: “ 5819, 6339, 5981, 4766, 5976”

Ours

    Input: “7 1 8 , 8 1 0 , 7 9 7 , 8 0 1 , 8 2 3 , 8 2 5 , 8 1 3 , 8 1 9 , 8 1 1 , 8 8 2 , … , 7 8 9 ,  6 1 8 ,  5 4 0 , 6 4 7 , 7 0 1”
    Output: “ 6 5 7 , 7 4 6 , 6 8 5 , 4 7 7 , 6 8 4 ”

PromptCast takes raw numerical values and inserts them into a question prompt (which is designed for a given dataset) then samples an answer. Our method rescales the numerical values and then directly extends the sequence of digits–without any natural language prompting. In our experiments, we found that scaling numbers before serialization and careful string formatting were absolutely necessary for good performance, while adding additional prompting was not necessary or helpful.

**Differences in performance**

To show how important our modeling choices were, we directly compare PromptCast’s string formatting method with our own. We use the DARTS datasets and the PISA + Multistep prompt format from the official PromptCast repo (which can be found under the path PISA/Dataset/PISA_Plus_Multistep_examples/*). We run sampling with both GPT-3 and LLaMA 70B. As PromptCast does not provide a method for calculating negative log likelihoods, we report CRPS and MAE values for each method.

| Model | CRPS | MAE |
|:-----------------|---------:|---------:|
| Ours (GPT-3) | 0.108 | 0.135 |
| Ours (LLaMA70B) | 0.104 | 0.142 |
| ARIMA | 0.113 | 0.125 |
| N-HiTS | 0.122 | 0.156 |
| TCN | 0.140 | 0.175 |
| SM-GP | 0.148 | 0.180 |
| N-BEATS | 0.170 | 0.210 |
| PromptCast (GPT-3) | 0.174 | 0.238 |
| PromptCast (LLaMA70B) | 0.171 | 0.234 |

While our method is best or second-best among all methods on both metrics, PromptCast is the worst overall. We think the most likely explanations for PromptCast's relatively poor performance are the effects of scaling and inconsistent tokenization.

**Approach to evaluation**

Beyond methodological differences and a large gap in practical performance, evaluation is also handled very differently in our work and in PromptCast. PromptCast’s metrics are solely deterministic and thus cannot capture ubiquitous stochasticity in time series data. Deterministic metrics can be problematic in time series analysis, leading to naive forecasts (e.g. simply predicting the last value in the input) obtaining state-of-the-art numbers [2]. For precisely this reason, we focused on probabilistic evaluations in the form of negative log likelihood (NLL) and continuous ranked probability score (CRPS). In addition to having only deterministic evaluations, PromptCast also provides much weaker evidence of generalization, with fewer datasets (3 vs. 20+) and no experiments disproving potential data leakage (memorization of test data by the LLM). Beyond basic forecasting evaluation, our analysis of LLMs’ favorable inductive biases (i.e. for deterministic extrapolation and uncertainty representations) and extra abilities (e.g. question answering) are also unique--not present in PromptCast or any other work we are aware of.

We thank the reviewers for their thoughtful comments. Addressing these concerns has already made our work stronger and its unique contributions more clear.

[1] Xue, Hao et al. "PromptCast: A New Prompt-based Learning Paradigm for Time Series Forecasting." 2022.

[2] Hansika Hewamalage et al. Forecast evaluation scientists: common pitfalls and best practices. Data Min Knowl Discov, 2023.

---

### Decision · Program_Chairs · 2023-09-21

**Decision:**

Accept (poster)

**Comment:**

This paper was a borderline paper and judging based on scores alone this paper should be rejected. However, I took the chance to read some of the review comments and also the paper itself. Skimming the paper, I find the paper to be of potential value to the community and I think it's interesting enough to be accepted especially with the impact multiplier of LLMs. I personally would be interested to read this paper so this is worth acceptance.

I also noticed the reviewer comments about not liking the attitude of the paper, which initially I thought could be a case of the authors "overclaiming". However, taking a look at the paper itself I find that that is not the case. In fact, I thought it was quite a tasteful presentation of the research questions.

Therefore, I recommend acceptance.